# Evaluation of Copernicus DEM and Comparison to the DEM Used for Landsat Collection-2 Processing

**Shannon Franks * and Rajagopalan Rengarajan** 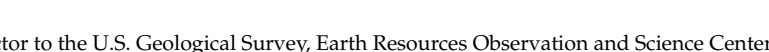

KBR, Contractor to the U.S. Geological Survey, Earth Resources Observation and Science Center, Sioux Falls, SD 57030, USA

\* Correspondence: shannonfranks@contractor.usgs.gov; Tel.: +301-614-6682

**Abstract:** Having highly accurate and reliable Digital Elevation Models (DEMs) of the Earth's surface is critical to orthorectify Landsat imagery. Without such accuracy, pixel locations reported in the data are difficult to assure as accurate, especially in more mountainous landscapes, where the orthorectification process is the most challenging. To this end, the Landsat Calibration and Validation Team (Cal/Val) compared the Copernicus DEM (CopDEM) to the DEM that is currently used in Collection-2 processing (called "Collection-2 DEM"). NGS ground-surveyed and lidar-based ICESat-2 points were used, and the CopDEM shows improvement to be less than 1 m globally, except in Asia where the accuracy and resolution of the DEM were greater for the CopDEM compared to the Collection-2 DEM. Along with slightly improved accuracy, the CopDEM showed more consistent results globally due to its virtually seamless source and consistent creation methods throughout the dataset. While CopDEM is virtually seamless, having greater than 99% of their data coming from a single source (Tandem-X), there are significantly more voids in the higher elevations which were mostly filled with SRTM derivatives. The accuracy of the CopDEM fill imagery was also compared to the Collection-2 DEM and the results were very similar, showing that the choice of fill imagery used by CopDEM was appropriate. A qualitative assessment using terrain-corrected products processed with different DEMs and viewing them as anaglyphs to evaluate the DEMs proved useful for assessing orbital path co-registration. While the superiority of the CopDEM was not shown to be definitive by the qualitative method for many of the regions assessed, the CopDEM showed a clear advantage in Northern Russia, where the Collection-2 DEM uses some of the oldest and least accurate datasets in the compilation of the Collection-2 DEM. This paper presents results from the comparison study, along with the justification for proceeding with using the Copernicus DEM in future Landsat processing. As of this writing, the Copernicus DEM is planned to be used in Collection-3 processing, which is anticipated to be released no earlier than 2025.

**Keywords:** orthorectification; DEM; Landsat; Copernicus; SRTM; ArcticDEM; NASADEM; Collection-3 processing; Collection-2; vertical accuracy



## 1. Introduction

The U.S. Geological Survey (USGS) Earth Resources Observation and Science (EROS) Center is responsible for processing and delivering all Landsat terrain-corrected Level-2 (surface reflectance) products to the public. Since 2016, they have performed so in a systematic manner called "Collections", whereas when there is a significant improvement to the radiometric and geometric accuracies used to process the data, the entire archive gets reprocessed to the latest standards of that Collection. The first implementation of this strategy was started in 2016, with the release of Collection-1. The products in Collection-1 used the Global Land Survey Digital Elevation Model (GLSDEM) for terrain correction of the imagery [1]. This Digital Elevation Model (DEM) is a mosaic of various elevation sources with varying levels of accuracy, quality, and resolution. The DEM was initially developed by MacDonald, Dettwiler, and Associates Federal (a subsidiary of Maxar Technologies)

in 2007 and later updated and renamed Collection-1 by the Calibration and Validation Team (Cal/Val) at USGS EROS. The Collection-1 DEM used the USGS National Elevation Dataset (NED) over the United States and Alaska, whereas Canadian Digital Elevation Data (CDED) were used over Canada. North of 60°N. latitude, a combination of Digital Terrain Elevation Data (DTED) level-1 and Global Multiresolution Terrain Elevation Data 2010 (GMTED2010) were used. For the rest of the landmass, the Consultative Group on International Agricultural Research (CGIAR) "hole-filled" Shuttle Radar Topography Mission (SRTM) data were used (see Figure 1 for the Collection-1 source DEM map). Since the original 2007 release, there were several improvements made to the Collection-1 SRTM data, which mainly included filling voids and removing artifacts [2].

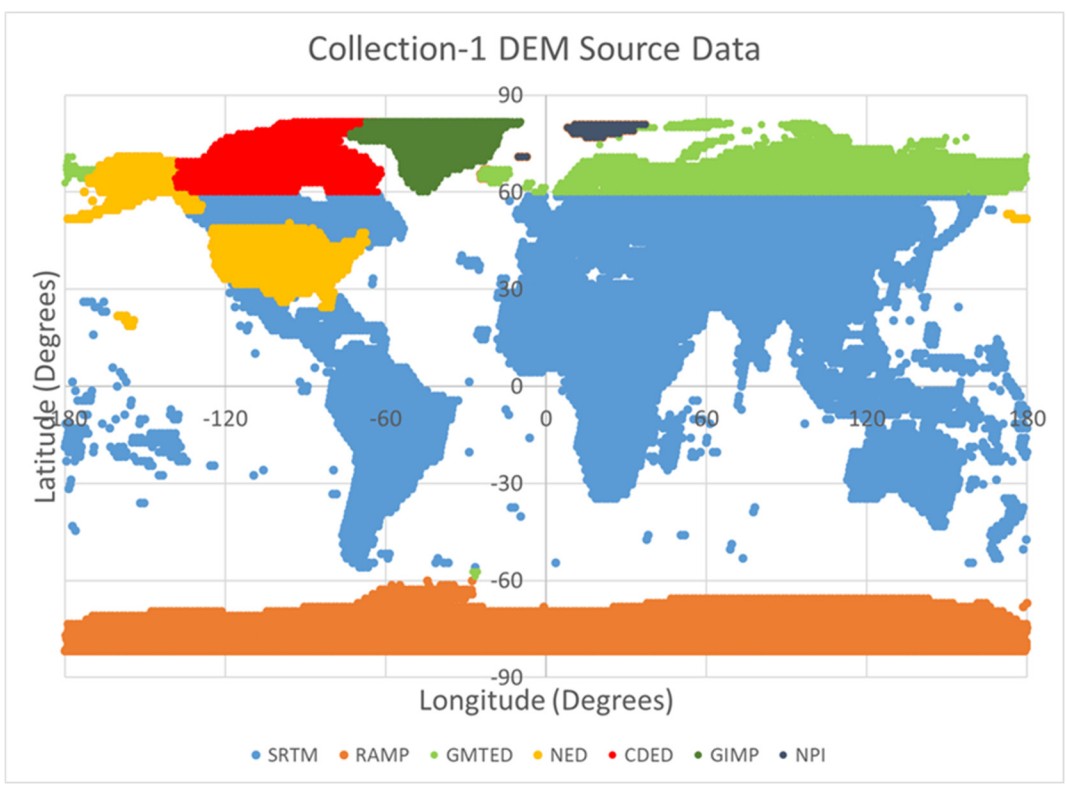

**Figure 1.** Landsat Collection-1 Digital Elevation Model (DEM) source data: Shuttle Radar Topography Mission [3], the Radarsat Antarctic Mapping Project (RAMP) [4], Global Multiresolution Terrain Elevation Data 2010 (GMTED2010) [5], National Elevation Dataset (NED) [6], Canadian Digital Elevation Data (CDED) [7], Greenland Icesheet Mapping Project (GIMP) [8], and NPI (Norwegian Polar Institute) [9].

Collection-2, which was released at the beginning of 2021, marked the second major reprocessing effort of the Landsat archive by the USGS [1]. There were many improvements in Collection-2 [10,11], among them an update to the global DEM used to orthorectify the data. At this point, the scientific community had access to DEMs with complete global coverage, better horizontal resolution, and improved relative and absolute vertical accuracies from products, such as AW3D30, Tandem-X DEM, and WorldDEM [12,13]. However, these datasets either were not global or had license restrictions that conflicted with the current USGS policy of free reference data distribution. As such, the Cal/Val Team chose to use freely available national and regional datasets. Collection-2 did not change the post spacing from Collection-1, maintaining it at 3 arcseconds (which equates to 90 m resolution at the equator), see Figure 2. It should be noted that in this paper, the term "resolution" is often used interchangeably with "post spacing" or "pixel size". To be consistent with other literature, we will continue the trend of using "resolution" as it is

often thought to be equivalent to elevation sample spacing, but this is not the most correct usage.

**Figure 2.** Landsat Collection-2 Digital Elevation Model (DEM) source map: Alaska-National Elevation Dataset (AK_NED) [6], Canadian DEM (CDEM) [7], NASADEM [14], Greenland Icesheet Mapping Project (GIMP) [8], NPI (Norwegian Polar Institute) [9], Sweden–Norway–Finland (SNF) [15–17], Global Multiresolution Terrain Elevation Data 2010 (GMTED2010) [5], the Radarsat Antarctic Mapping Project (RAMP) DEM in Antarctica [4], and ArcticDEM [18].

Even though the Collection-2 DEM was an improvement over the Collection-1 DEM, there were still some drawbacks which motivated the Cal/Val team to further improve its next rendition of the collections' dataset: (1) there were three regions where the DEMs were not updated (Greenland, Antarctica, and most of northern Russia), (2) even with the increased accuracy, some datasets had errors induced by void filling (ArcticDEM), (3) the dates when these datasets were collected varied greatly so the user needed to be careful when applying them to more current land cover changes, and (4) although hole-filled, in some of the highest elevations, NASADEM still had many artifacts. As such, after the completion of the Collection-2 dataset, there were two major goals for the next rendition of the Collection series: (1) use a globally "complete" dataset so the collection and production methods are uniform throughout (i.e., quality and accuracy are not patchworks) and (2) improve the spatial resolution to 1 arcsecond (~30 m) to detect the sharper elevation changes in mountainous terrains. The Copernicus DEM datasets (release 1) were made available in December 2019, about a year before the USGS started using the Collection-2 DEM in data processing. However, the first release was only of the 90 m product which was not of great interest to the team. A year after that, in November 2020, the 30 m product was released [19], but the USGS Cal/Val team needed time to evaluate it before implementing it into processing and did not want to delay the release of Collection-2. Therefore, it was planned to evaluate the dataset and consider it for Collection-3 processing if the quality was indeed superior. Additionally, since there were three Copernicus DEM releases with updates in each version, it was decided to wait until the final release before any analysis was executed.

Researchers have studied the accuracy and consistency of different DEM datasets over the last several decades. Some of their analysis have focused on (a) different metrics that are useful in the comparison and evaluation of the DEM datasets [3,20], (b) analysis using derived products such as slope and shaded-relief products [21,22], (c) relative comparison of different DEM datasets [20,23,24], (d) analysis of the DEM datasets based on applications [25–27], (e) comparison of DEM datasets using Lidar point cloud and other high-resolution locally surveyed points [28]. In our study, we have focused our analysis

primarily on the quantitative and qualitative comparisons between the Copernicus DEM and Collection-2 DEM. We have not only used some of the metrics established by the above-cited researchers for quantitative evaluation but have also introduced a unique qualitative comparison methodology that is independent of the need for reference datasets.

The Copernicus DEM is provided at three resolutions [29], with the 1 arcsecond (1″) version, here called CopDEM, being the one that the USGS Cal/Val team has evaluated. The 1″ and 3″ datasets cover the full global landmass (see Figure 3) with the timeframe of data acquisition between 2010 and 2015.

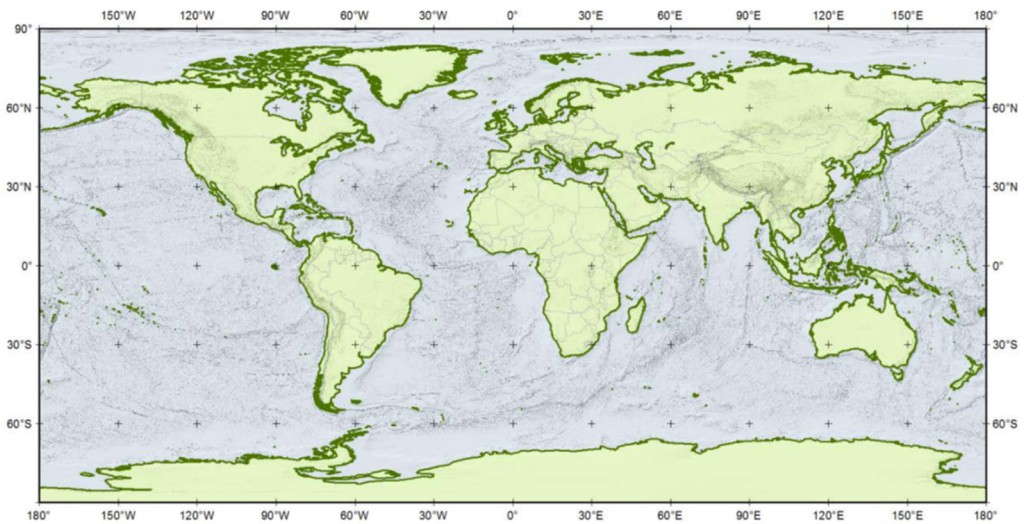

**Figure 3.** Copernicus Digital Elevation Model (DEM) coverage.

This paper describes the methods and results from the qualitative and quantitative comparison between the CopDEM and the Collection-2 DEM. This paper is structured as follows. First, the datasets that were evaluated are introduced. Then, in the methodology section (Section 3), we describe how each of the evaluated datasets was processed, how the reference datasets were selected and used, and how anaglyphs were created to determine misalignment when viewed at different cross-track angles. In Section 4, the results and discussion of the study are given, and, lastly, in Section 5, conclusions are presented summarizing the results, improvements, and limitations of the new "Collection-3" DEM.

## 2. Datasets Used in the Study

### 2.1. Copernicus DEM (CopDEM)

#### 2.1.1. Primary Motivation and Usage [30,31]

Copernicus is the name of the European Union's Earth observation program coordinated and managed by the European Commission in partnership with the European Space Agency (ESA), the EU Member States, and EU agencies. The CopDEM dataset suite was created to further advance the harmonization of spatiotemporal data within the Copernicus Program. This globally homogenous DEM provides the Copernicus user community with an improved and harmonized high-quality dataset with the goal of achieving a global, continuous, autonomous, high-quality, wide-range Earth observation capacity. An important purpose for the CopDEM is the orthorectification of Sentinel-2 data products, providing a highly accurate and consistent DEM.

#### 2.1.2. Heritage

Over the last few decades, synthetic aperture radar (SAR) interferometry (InSAR) and, more recently, high-resolution stereo methods have become the standard methods for obtaining DEMs at a global scale [32–34]. The most recent SAR mission is TanDEM-X, which set out to become a successor to SRTM [13,35–37] and collected its data between 2010

and 2015 [35]. The TanDEM-X mission was a constellation of two satellites (TerraSAR-X and TanDEM-X), with the primary mission goal to generate a global DEM. DLR produced three DEMs at resolutions of 12 m (0.4 arcseconds), 30 m (1 arcsecond), and 90 m (3 arcseconds), respectively. These DEMs are unedited versions resulting from interferometric processing and mosaicking only [38] and contain artifacts, such as voids, spikes, and holes. The TanDEM-X DEM was first edited by Airbus Defense and Space and made commercially available as WorldDEMTM [39]. In addition to terrain and hydrology editing, they created three different products: a version including the pits, spikes, and voids called as WorldDEMcore, an edited Digital Surface Model called as WorldDEM, and the last product being an edited Digital Terrain Model called as WorldDEM DTM. Meanwhile, with the objective of procuring a global and consistent high-resolution DEM for usage within the Copernicus program, following the open tender by ESA, selected WorldDEM as the basis for the Copernicus DEM products [40]. WorldDEM was used to produce three Copernicus DEMs: EEA–10 (0.4 arcseconds), GLO–30 (1.0 arcseconds), and GLO–90 (3.0 arcseconds), with the former (EEA–10) only available over wider European countries (the so-called "EEA 39") and the latter two are available as a global DEM product.

### 2.1.3. Format and Quality Layers

The CopDEM data are available in three different formats, each having different specifications.

| | |
|---|---|
| DGED format: | EEA [1]−10, GLO [2]−30, GLO–90 |
| DTED format: | GLO–30, GLO–90 |
| INSPIRE format: | EEA–10 |

[1] European Environmental Agency, at 10 m resolution. [2] Global, at 30 or 90 m resolution.

Defense Gridded Elevation Data (DGED) formatted CopDEM data cover all the resolutions that have been produced. DGED 30 and 90 m data are global and provided as 32-bit floating point data in GeoTIFF file format including the corresponding XML metadata and quality layers. Additionally, it is the DGED format in which the original TanDEM-X data are disseminated. For this study, we used the GLO–30 DGED formatted data so we can use their quality layers, which were not available with the other formats.

The Digital Terrain Elevation Data (DTED) version of the CopDEM is also global but is provided as 16-bit signed integer in GeoTIFF format. Corresponding XML files are provided with this version, but no quality layers. This version was not used in this analysis but will be used for product generation by the USGS since the file size is smaller and the additional bit of precision only makes a marginal difference, if any, when processing the terrain correction at Landsat field of view and resolutions.

Each Copernicus tile was provided in $1° \times 1°$ tile size with variable longitudinal grid size depending on the latitude (see Table 1 for DGED resolution reduction specifications and Figure 4 for an overview map [29]). The vertical unit for measurement of elevation height is meters, and all Copernicus data are referenced to the EGM08 geoid. As noted above, the primary reason for doing the analysis using the DGED version of files is that along with the elevation GeoTIFFs, six quality layers also come with the data. These quality layers helped us to understand the data and explain some of the results (see Table 2). The quality layers most relevant to this study were the Editing Mask, which showed if individual pixels in GeoTiff DEM image are original CopDEM-derived pixels or have been edited/adjusted, and the Filling Mask, which showed what source DEMs were used to fill pixels (see Table 3 for Edit Mask layer and Table 4 for Fill Mask layer). The combination of these two layers was helpful in explaining trends and errors that were observed in the data.

### 2.1.4. Copernicus DEM Accuracy

Like all datasets, DEM accuracy varies per land cover type. Official statistics published by the Copernicus program state that overall absolute vertical accuracy at a 90% (LE90)

confidence level is <4 m [29]. Almost 95% of the 1° tiles have an absolute vertical accuracy value better than 3 m and around 2% of the 1° tiles have an accuracy value greater than 5 m, which are clustered in mountainous terrain in forested canopy [29]. Due to perpetual ice cover, Greenland and Antarctica were separated from the analysis, but the accuracy is slightly worse than 5 m [29].

**Table 1. Copernicus Digital Elevation Model** (CopDEM) grid spacing and longitudinal reduction factors following Defense Gridded Data (DGED) format. Copernicus DEMs: European Environmental Agency EEA–10 (0.4 arcseconds), GLO–30 (1.0 arcseconds), and GLO–90 (3.0 arcseconds).

| | | DGED Format | | | | | |
|---|---|---|---|---|---|---|---|
| **Copernicus DEM instance** | | EEA–10 | | GLO–30 | | GLO–90 | |
| LAT spacing | | | | | | | |
| LON spacing | 0–50° | 0.4″ | 1× | 1.0″ | 1× | 3.0″ | 1× |
| | 50–60° | 0.6″ | 1.5× | 1.5″ | 1.5× | 4.5″ | 1.5× |
| | 60–70° | 0.8″ | 2× | 2.0″ | 2× | 6.0″ | 2× |
| | 70–75° | 1.2″ | 3× | 3.0″ | 3× | 9.0″ | 3× |
| | 75–80° | | | | | | |
| | 80–85° | 2.0″ | 5× | 5.0″ | 5× | 15.0″ | 5× |
| | 85–90° | 4.0″ | 10× | 10.0″ | 10× | 30.0″ | 10× |

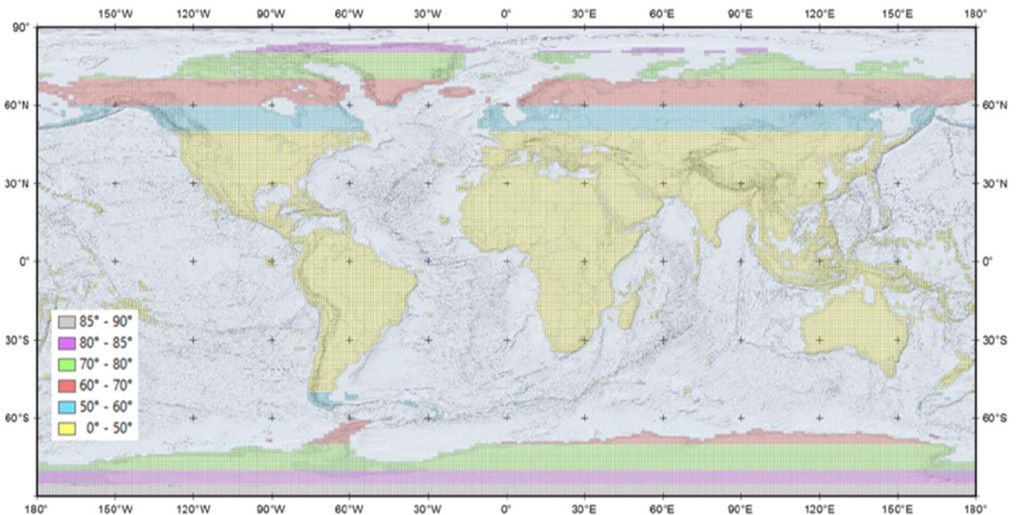

**Figure 4.** Map of Copernicus latitudinal reduction factor [29].

**Table 2.** Copernicus Digital Elevation Model (CopDEM) quality layers.

| | Quality Layers | Data Format |
|---|---|---|
| Editing Mask | EDM | 8-bit unsigned integer, GeoTIFF |
| Filling Mask | FLM | 8-bit unsigned integer, GeoTIFF |
| Height Error Mask | HEM | 32-bit floating point, GeoTIFF |
| Water Body Mask | WBM | 8-bit unsigned integer, GeoTIFF |
| Source Data Layer | SRC | KML vector file |
| Accuracy Layer | ACM | KML vector file |

**Table 3.** Copernicus Edit Mask Layer.

| Pixel Value | Meaning |
|:---:|:---|
| 0 | Void (no data) |
| 1 | Not edited |
| 2 | Infill of external elevation data |
| 3 | Interpolated pixels |
| 4 | Smoothed pixels |
| 5 | Airport editing |
| 6 | Raised negative elevation pixels |
| 7 | Flattened pixels |
| 8 | Ocean pixels |
| 9 | Lake pixels |
| 10 | River pixels |
| 11 | Shoreline pixels |
| 12 | Morphed pixels (series of pixels manually set) |
| 13 | Shifted pixels |

**Table 4.** Copernicus Fill Mask Layer.

| Pixel Value | Meaning |
|:---:|:---|
| 0 | Void (no data) |
| 1 | Edited (except filled pixels) |
| 2 | Not edited/not filled |
| 3 | ASTER [2] |
| 4 | SRTM90 [3] |
| 5 | SRTM30 [3] |
| 6 | GMTED2010 [4] |
| 7 | SRTM30plus [5] |
| 8 | TerraSAR-X Radargrammetric DEM |
| 9 | AW3D30 [6] |

[2] Advanced Spaceborne Thermal Emission and Reflection Radiometer (ASTER) Global Digital Elevation Map retrieved from https://asterweb.jpl.nasa.gov/gdem.asp (accessed on 2 May 2023). [3] Shuttle Radar Topography Mission (SRTM) Digital Elevation Data retrieved from http://earthexplorer.usgs.gov/ (accessed on 2 May 2023). [4] GMTED2010 Elevation Data retrieved from http://earthexplorer.usgs.gov/produced by the U.S. Geological Survey, https://lta.cr.usgs.gov/sites/default/files/Data%20Citation_1.pdf (accessed on 2 May 2023). [5] National Aeronautics and Space Administration Land Processes Distributed Active Archive Center (NASA LP DAAC), 2013, NASA Shuttle Radar Topography Mission Global 1 arc second, Version 3.0. NASA EOSDIS Land Processes DAAC, 2013 USGS Earth Resources Observation and Science (EROS) Center, Sioux Falls, South Dakota (https://lpdaac.usgs.gov (accessed on 2 May 2023)), accessed May 2nd 2017 at https://doi.org/10.5067/MEaSUREs/SRTM/SRTMGL1.003 (accessed on 2 May 2023). [6] ALOS World 3D-30m (AW3D30) provided by Japan Aerospace Exploration Agency (JAXA) retrieved from https://www.eorc.jaxa.jp/ALOS/en/aw3d30/data/index.htm (accessed on 2 May 2023).

For additional details on the CopDEM, refer to the Copernicus Product Handbook [29].

*2.2. Landsat Collection-2 DEM*

Since the beginning of 2021, the Landsat program has been using the Collection-2 DEM (https://www.usgs.gov/landsat-missions/landsat-collection-2-digital-elevation-model (accessed on 2 May 2023) and can be downloaded from https://earthexplorer.usgs.gov) for terrain correction in its product generation system. The accuracy of this dataset was a big improvement over the previous, Collection-1 DEM that was being used showing differences in some places (i.e., where ArcticDEM was used) of up to 35 m [1]. The source data that comprise the Collection-2 DEM are the Canadian DEM (CDEM) [7], Greenland Icesheet Mapping Project (GIMP) [8], Sweden–Norway–Finland (SNF) [15–17], Alaska-National Elevation Dataset (AK_NED), Global Multiresolution Terrain Elevation Data 2010 (GMTED2010), NPI (Norwegian Polar Institute) [9], ArcticDEM, the Radarsat Antarctic Mapping Project (RAMP) DEM in Antarctica, and for most of the globe, NASADEM (see Figure 2). NASADEM improved upon SRTM by filling voids, removing large artifacts, and by reprocessing the original SRTM data using new software and ancillary data that did

not exist in the original processing [2]. Since the Collection-2 DEM used many different source DEMs, there was a considerable amount of processing that was completed to make the overall dataset seamless. Results, however, showed that on average, improvements to absolute vertical accuracies were better than 35 m (STD) in places of Northern Eurasia that used ArcticDEM and horizontal improvements were noted in some of the most mountainous regions to be in the range of multiple 30 m Landsat pixels [1]. Similar to the Collection-1 DEM, the resultant Collection-2 dataset was produced at 3 arcseconds, was vertically referenced to the Earth Gravity Model of 1996 (EGM96) Geoid, and models the surface elevation (i.e., Digital Surface Model—DSM). For details on each of the source DEM specifications, including pixel size, vertical reference, published accuracies, and processing steps, refer to [1].

### 2.3. National Geodetic Survey (NGS)

In North America, National Oceanic and Atmospheric Administration (NOAA) NGS survey points were used to estimate the DEM accuracies. These GPS "benchmarks" have been crowd-sourced since 2014 and provide extremely high absolute accuracy, with a specification of $\pm 1$ cm $(1-\sigma)$ on the ellipsoid heights of the points [41]. The specific model that was used was the GEOID18 points [42], which is the NGS's latest hybrid geoid model and is a significant improvement over its predecessor, having a lower standard deviation of error while adding 29% more GPS on benchmark observations, increasing the total to over 32,000 in the continent. For comparison to the Collection-2 DEM, the NGS points were relative to the EGM96 geoid, and for comparison to the CopDEM, the points were converted to the EGM08 geoid using NOAA-hosted vertical datum transformation tool (VDatum).

### 2.4. Ice, Cloud, and Land Elevation Satellite (ICESat)

The Ice, Cloud, and land Elevation Satellite-2 (ICESat-2) is the second generation of the laser altimeter ICESat mission and was launched in 2018 [43]. The specific product used for our analysis was ATL08, which provides land and vegetation height at a given point on the Earth's surface at a given time relative to the World Geodetic System of 1984 (WGS-84) ellipsoid, which was converted to heights relative to the geoid using VDatum tool [44]. ATL08 provides estimates of the surface height, along with ancillary parameters needed to interpret and assess the quality of these height estimates. ICESat-2 data have better precision and accuracy than ICESat-1, and studies have shown the Root Mean Square (RMS) error between the airborne Lidar and ICESat-2 sensors is better than 2 m. However, it is important to note that the elevation error in ICESat data increases with increasing slope and vegetation cover [45–47].

## 3. Methodology

For this project, our task was to evaluate the CopDEM both quantitatively and qualitatively in relation to the Collection-2 DEM that is currently being used in Landsat processing. The methodology followed had three parts: (1) for North America, use NGS benchmark points to determine the datasets' accuracy, (2) for the rest of the globe, ascertain the datasets' accuracy by comparing them to extracted elevation points collected by the ICESat-2 sensor, and (3) create Landsat-orthorectified anaglyphs of specific regions using both DEM datasets to determine terrain parallax-induced errors from adjacent collects along the Worldwide Reference System (WRS-2).

### 3.1. Accuracy Assessment

#### 3.1.1. NGS Points

National Geodetic Survey (NGS) points [48] were used to ascertain the absolute vertical accuracy of the datasets. These orthometric height measurements were downloaded from the NOAA NGS website. The data are primarily from the United States; however, some points in Canada and Mexico were available and used (see Figure 5). Using GIS software,

the NGS point locations were extracted using bilinear interpolation from the DEM raster images and differenced with the NGS point's height (NGS point heights minus DEM height) to calculate accuracy. To ensure that the vertical datums matched, the NGS points that were extracted from the CopDEM were compared with the NGS points referenced to the EGM08 geoid, and the ones extracted from the Collection-2 DEM were compared with the NGS points referenced to the EGM96 geoid.

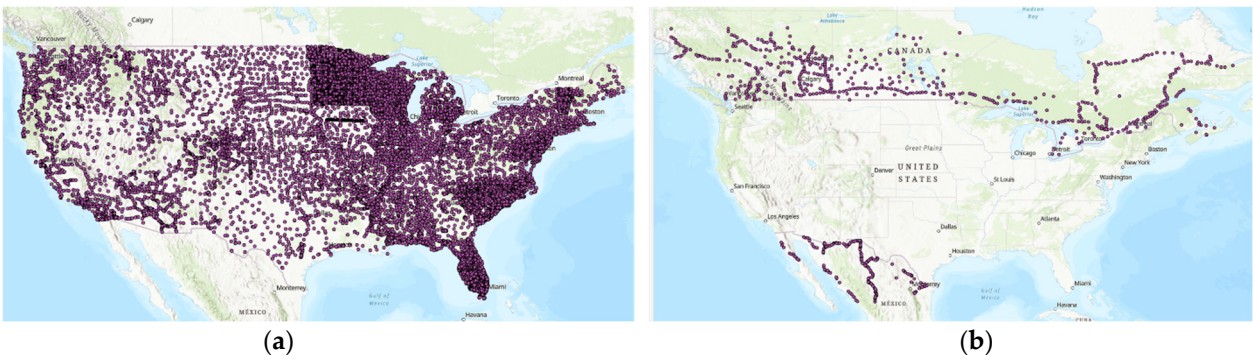

(**a**)                                                                          (**b**)

**Figure 5.** NGS point distribution in (**a**) the United States and (**b**) Canada/Mexico.

### 3.1.2. ICESat-2 Advanced Topographic Laser Altimeter System (ATLAS) Points

Expanding on the North America accuracy assessment using the NGS points, ICESat-2 ATLAS data were used to obtain global statistics. A sampling approach was used, where 7–18 study areas per continent were selected based on topography, geographical representation, and the Collection-2 DEM source data type. A total of 60 sites (Figure 6) were selected, with each site being 1° × 1° tile size, and having an average of 15,000 ICESat-2 points in each. While this strategy is not sufficient to perform a robust evaluation of the datasets' global accuracy, it allowed us to evaluate if the CopDEM is more accurate than the Collection-2 DEM in some high-relief and challenging terrain regions.

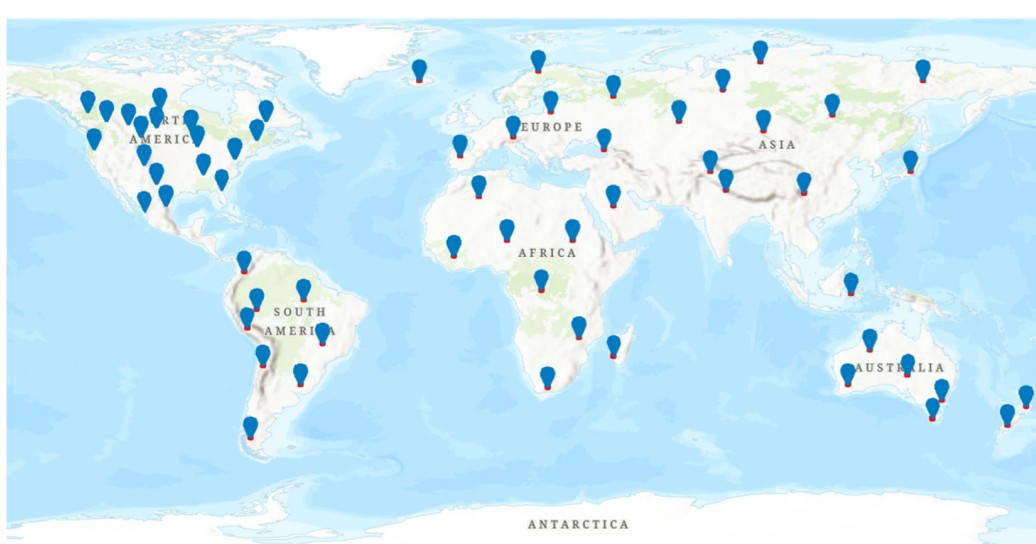

**Figure 6.** Site distribution for ICESat-2 analysis.

From NASA's Earthdata website [49], ATL08 points were filtered based on certain thresholds using metadata information available and downloaded with the ATL08 product. Using these attribute bands, the data were filtered to ensure that only high-quality points were used in the analysis. Following suggestions from other research [50–52], the data were screened to have low uncertainty (h_te_uncertainty < 10 m), a low standard deviation (h_te_STD < 4), a high number of photon returns (n_te_photons > 50), a low point-spread

function (PSF_flag = 0), and closeness to a reference DEM (DEM_removal_flag = 0). This typically eliminated roughly 50% of the observations. After the filtering, a few tiles on coastlines still had points over the oceans which were additionally eliminated. The h_te_best_fit parameter provided in the ATL08 metadata was used as the ellipsoidal height for the corresponding horizontal coordinates. This height provides an estimate of the best-fit height of the ground photons at the center of each 100 m ground sampling distance (adjusted for local topography).

To match the vertical reference frame of the CopDEM (EGM08) and Collection-2 DEM (EGM96), the ellipsoidal height was converted to orthometric by applying the geoid separation values for each point using NOAA's VDatum software. The ICESat points were then imported into ESRI's GIS software for analysis, and the DEM heights were extracted via bilinear interpolation from the two raster DEM datasets for the corresponding horizontal locations of the ICESat points. The difference statistics, such as mean, standard deviation, range, and root mean square error (RMSE), were calculated (ICESat-2 point heights minus DEM heights) between the ICESat elevation and the corresponding DEM-interpolated elevations.

*3.2. Anaglyph Analysis*

DEM-to-DEM comparisons, as we did in this study, show differences but do not explicitly indicate accuracy. Using DEMs from adjacent paths (i.e., differing view angles) to terrain-correct Landsat data provides a different perspective on DEM accuracy because this approach is independent of any reference source that can potentially introduce error [53], and it incorporates the effect of the sensors' viewing geometry to translate if differences in vertical heights lead to better horizontal accuracy (assuming that the sensors have highly accurate point knowledge). Viewing DEMs as anaglyphs is a simple but effective way to convey depth perception and bring the third dimension onto a 2D screen essentially recreating how the human brain interprets depth: by viewing two images at slightly different angles [54,55]. Comparing imagery that is terrain-corrected using different DEMs shows errors in the form of misalignments when stacking the off-angle images. This is pictorially demonstrated in Figure 7, where a sensor views a target (peak) from different view angles ($\alpha$ and $\beta$). The line of sight (Sat$_1$) projects the target to the ellipsoid at (B$_{S1}$) and the line of sight (Sat$_2$) projects the same target at B$_{S2}$. If the DEM is accurate, and the sensor pointing knowledge is accurate, then the orthorectification process corrects the apparent line of sight to the actual horizontal location at point A. However, if the DEM is inaccurate then the two apparent lines of sight (B$_{S1}$ and B$_{S2}$) will be corrected in the orthorectification process to different locations (A$'_{S1}$ and A$'_{S2}$), respectively. Thus, the orthorectification process introduces horizontal misalignments ($\Delta d$) in the images from two different angles while viewing the same target. This is also explained in Equations (1–3), where the magnitude of the displacement ($\Delta d$) is directly proportional to the erroneous height differences, $\Delta h_A$, simplified by using a flat Earth model. Figure 8 shows an example of a satellite imaging an object on the ground at two varying angles creating scenes of an object (i.e., a mountain peak) in various spectral bands. Those images get processed with ancillary data including a DEM, for terrain correction. In our anaglyph analysis, we processed those images using both the Collection-2 DEM and the CopDEM and then created the anaglyphs using processed panchromatic bands from different view angles. Then the anaglyphs are compared visually for differences.

$$\Delta HT = (hA) \tan (|\text{view angle}|) \tag{1}$$

$$\Delta HA = (hA + \Delta hA) \tan (|\text{view angle}|) \tag{2}$$

$$\Delta d = \Delta hA \{\tan (|\text{view angle } \alpha|) + \tan (|\text{view angle } \beta|)\} \tag{3}$$

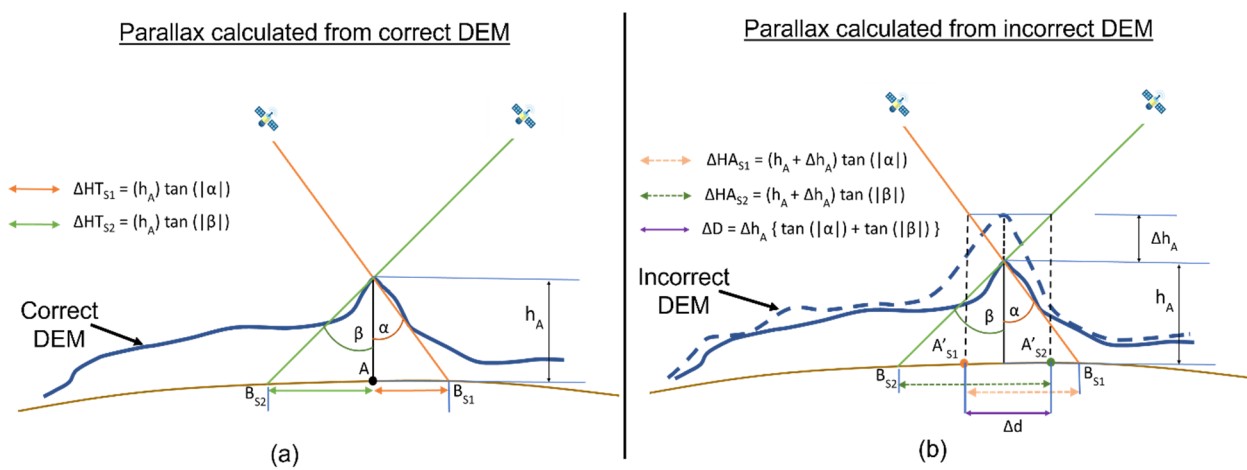

**Figure 7.** As seen in the left image (**a**), when there is no elevation error, the view angles ($\alpha$ and $\beta$) along with the elevation height ($h_A$) calculate two horizontal displacements ($\Delta HT_{S1}$ and $\Delta HT_{S2}$) that locate the points to the same horizontal coordinates, A. As seen in the right image (**b**), when calculating this based on an incorrect Digital Elevation Model (DEM) height, the displacements ($\Delta HA_{S1}$ and $\Delta HA_{S2}$) place the horizontal coordinates in differing locations ($A'_{S1}$ and $A'_{S2}$) causing misalignment ($\Delta d$).

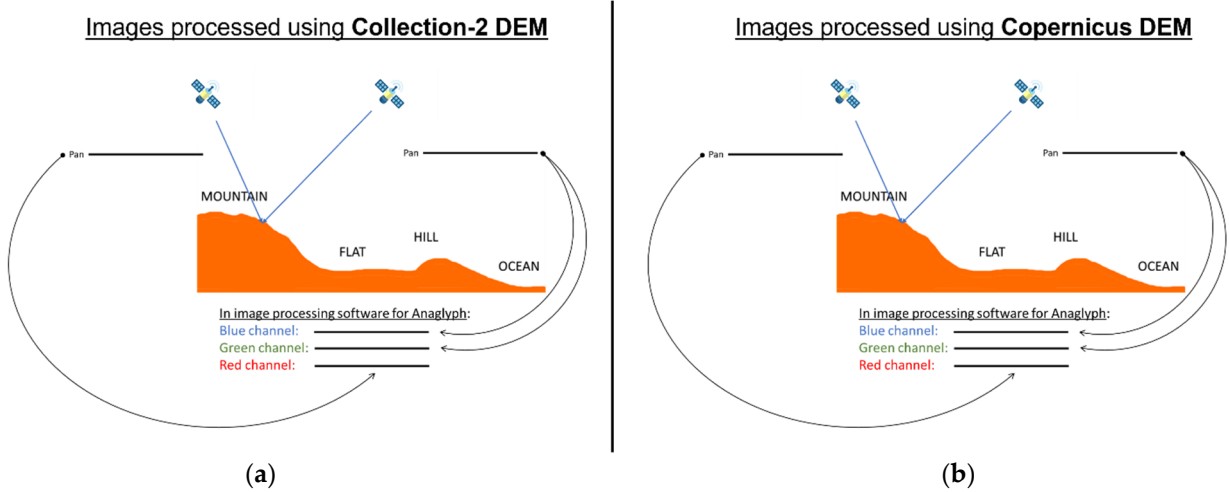

**Figure 8.** Illustration of parallax for Anaglyph creation (**a**) using Collection-2 Digital Elevation Model (DEM) and (**b**) the Copernicus DEM (CopDEM). The methodology is exactly the same in each, except that they use different DEMs for creation.

If the Region of Interest (ROI), such as a peak or valley, in the stacked images do not perfectly align, it will show in the anaglyph as a variation in color (a shift in Red, Green, or Blue) depending on which channels were used to view the individual image bands. When the Pan images from two different view angles align perfectly, there is no color shift and the RGB composite image is displayed as grayscale (i.e., an equal amount of Red, Green, and Blue). These shifts are best viewed using 3D glasses to increase depth perception and better highlight the parallax-induced error [56]. Note that the inaccuracy in the DEM height is better established and visualized only when the view angle differences are larger. In the case of nadir-viewing, any discrepancy in the height will not be visible. Similarly, the effect will be more pronounced for high terrain or terrain changes, as flat terrain at mean sea level (MSL) would introduce no significant differences. See Figure 9 for an example of two anaglyphs created using an Austrian national DEM and WorldDEM in the Austrian Alps.

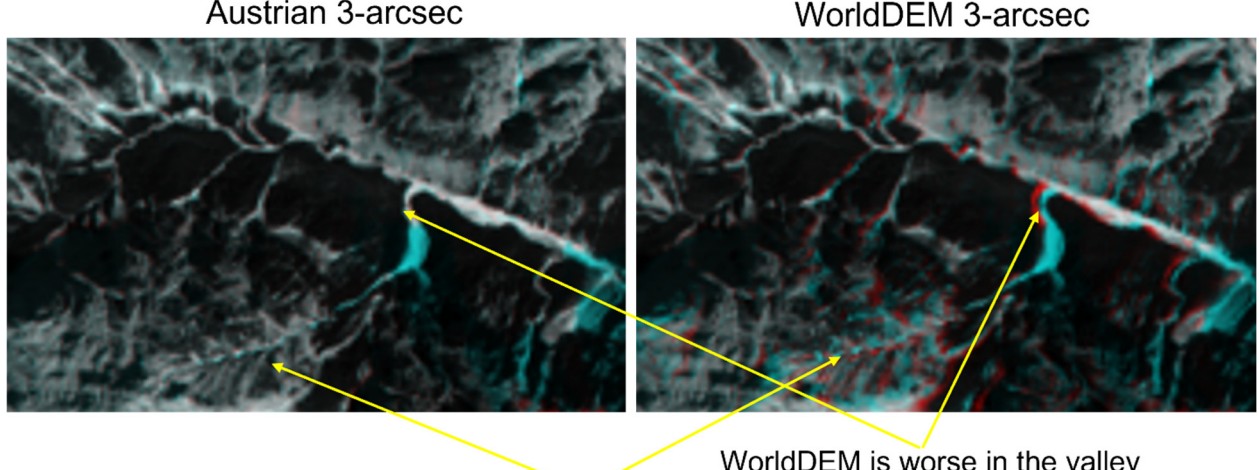

**Figure 9.** Anaglyph example in Austria using a national Digital Elevation Model (DEM) (from Austria) and WorldDEM, the precursor DEM to Copernicus DEM (CopDEM).

The steps for creating the anaglyphs were as follows:

1. Find a mountainous region that has significant path-to-path overlap in the WRS-2 grid.
2. Within those two overlapping grid cells, process two Landsat images to Level-1 Precision Terrain Correction (L1TP) with the CopDEM and then process them using the Collection-2 DEM. Acquisition dates for the two selected images must be very close (i.e., we found it best if the acquisition dates were within 10 days of one another) to minimize seasonal differences. We prioritized seasonal closeness over temporal closeness, but both were preferred if possible. We used Band 8 (panchromatic) from the Landsat 8 satellite to take advantage of the increased 15 m resolution. We used the USGS Image Assessment System (IAS) to process the imagery.
3. For each of the image pair composites, set the Green and Blue bands to the same scene and Red to be the other scene from a different view angle. Do this for both cases (with CopDEM and Collection-2 DEM) and compare the results to see if there are differences in the anaglyphs. If one of the RGB composites presents additional shifts in color patterns then there are misalignments between the two images used to create that composite, which are due to the elevation error in the DEM dataset.

### 3.2.1. South of 60°N. Latitude: NASADEM, Focusing on Regions of High Relief

South of 60° latitude, the focus was on creating anaglyphs in regions of high relief since these areas are where DEM inaccuracies are most prevalent and where the displacements would most likely occur. Additionally, regions of sharp terrain are where the increased resolution of the CopDEM should be most beneficial since greater resolution would improve the ability to capture the changing topography rather than have those features smoothed out, as would occur with lower-resolution data. To find the regions of high relief, we chose to use a slope map derived from SRTM data used in previous work (see Figure 10). The intersection of the slope map with the nominal WRS-2 geographic extent helped to identify the site location for this analysis. Note that two criteria were used in the site selection. First, the selected region should have a high terrain slope, and secondly, it should be at the edge of the Landsat field-of-view so there is overlap at the sides (see Figure 11 for an example of overlap in an area of high slope). Anaglyphs were created for three sites, one in the Austrian Alps near Innsbruck, another in the Himalayas along the China/Nepal border, and the third in the Northern Himalayas in Pakistan (see Figure 12 and Table 5 for the specifics of each region).

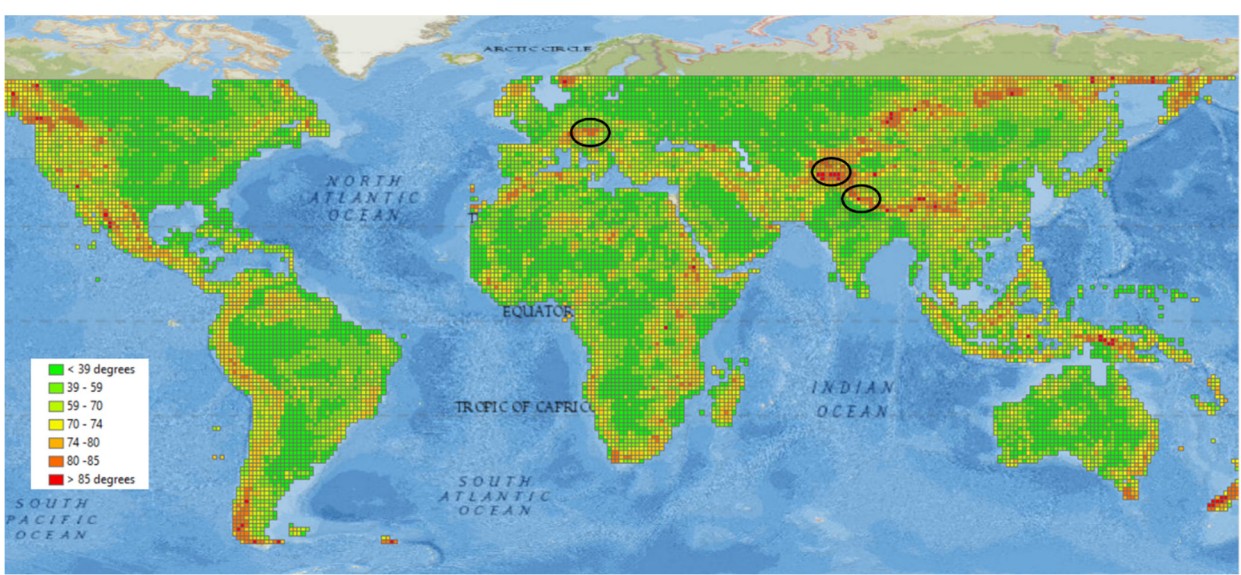

**Figure 10.** Shuttle Radar Topography Mission (SRTM) slope map showing locations of the study sites.

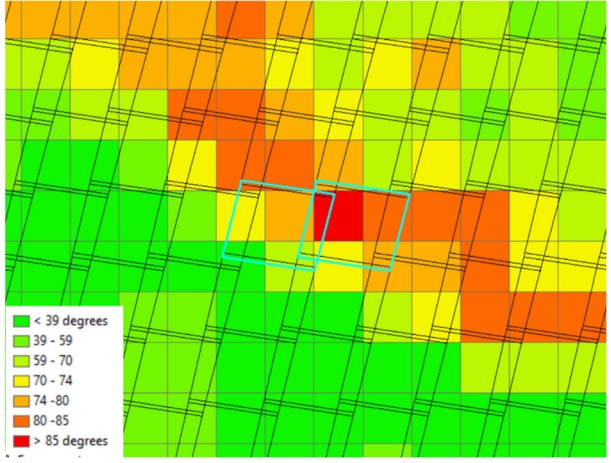

**Figure 11.** Landsat Worldwide Reference System−2 (WRS-2) overlap example.

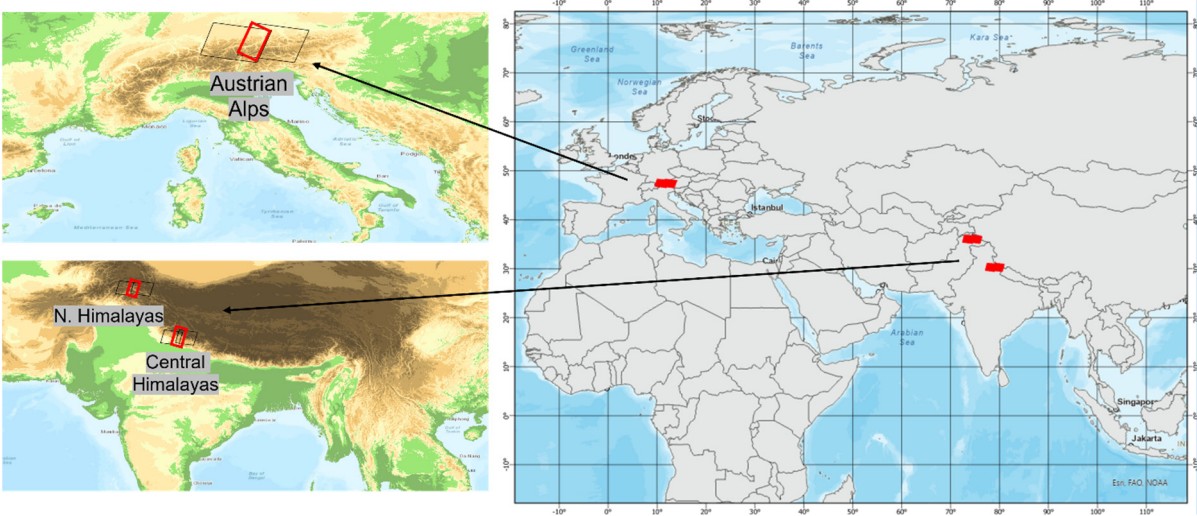

**Figure 12.** Anaglyph locations south of 60°.

**Table 5.** Landsat scene combinations used to create anaglyphs south of 60° latitude. WRS-2, Landsat Worldwide Reference System-2.

| Location | WRS-2 Combo | Scenes Used | Max Slope | Elevation Range |
|---|---|---|---|---|
| Austrian Alps | 192/27 and 193/27 | LC81920272019264LGN00 | 83° | 132–3978 m |
| | | LC81930272020258LGN00 | | |
| Himalayas (China/Nepal) | 145/39 and 146/39 | LC81460392015251LGN01 | 85° | 203–7804 m |
| | | LC81450392020274LGN00 | | |
| N. Himalayas (Pakistan) | 149/35 and 150/35 | LC81490352018216LGN00 | 86° | 591–8570 m |
| | | LC81500352018223LGN00 | | |

3.2.2. North of 60°N. Latitude: SNF, ArcticDEM, and GMTED

North of 60°N. latitude, we concentrated on locations where various other source DEMs were used in the Collection-2 dataset. In Scandinavia, where the SNF DEM was used in Collection-2 processing, various national datasets (i.e., one for Sweden, Norway, and Finland) were used, and we wanted to test how seamless the SNF DEM was along the Sweden–Norway border. In Iceland, where ArcticDEM was used, we knew that the accuracy of the dataset is generally good but there were a lot of voids in the DEM dataset that required filling, and we wanted to test if those voids show up in the anaglyph analysis. In Northern Russia, the source DEM was GMTED which was never updated in Collection-2. Since the GMTED is an old DEM dataset with known artifacts, we wanted to verify the errors using anaglyph analysis in this region (see Figure 13 for site locations and Table 6 for the specifications).

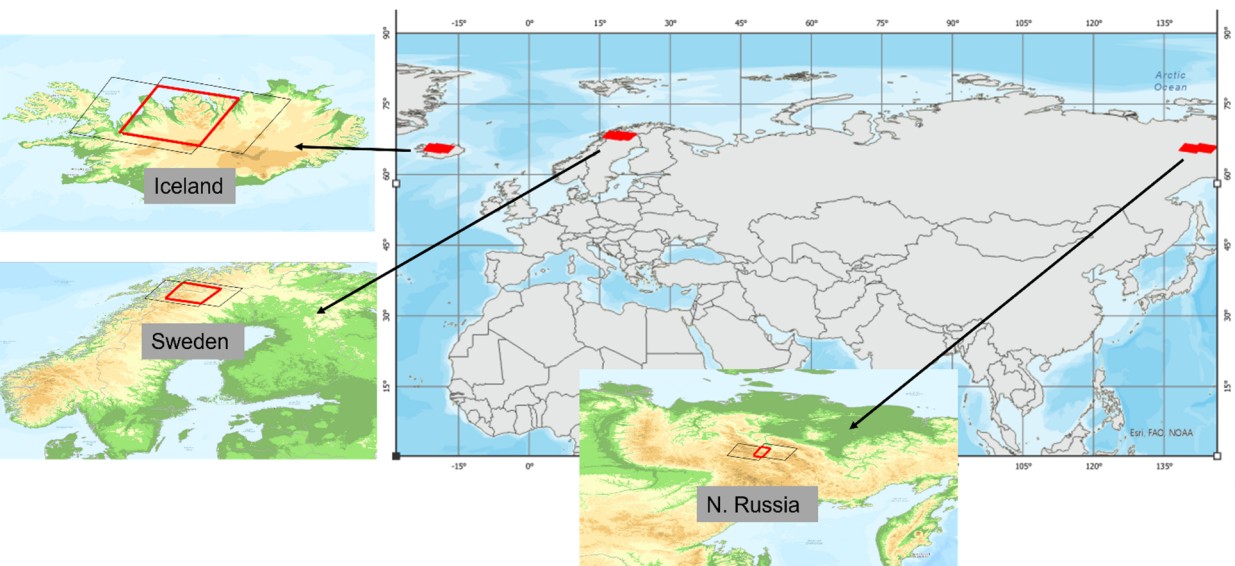

**Figure 13.** Anaglyph locations north of 60°.

**Table 6.** Landsat scene combinations used to create anaglyphs north of 60° latitude. WRS-2, Landsat Worldwide Reference System-2; SNF, Sweden–Norway–Finland; ArcticDEM, Arctic Digital Elevation Model; GMTED, Global Multiresolution Terrain Elevation Data.

| Location | WRS-2 Combo | Scenes Used | C-2 Source DEM |
|---|---|---|---|
| Sweden/Norway | 196/12 and 197/12 | LC81960122020247LGN00 | SNF |
| | | LC81970122021208LGN00 | |
| Iceland | 219/14 and 220/14 | LC82190142018258LGN00 | ArcticDEM |
| | | LC82200142018249LGN00 | |
| N. Russia | 114/14 and 116/14 | LC81140142015251LGN01 | GMTED |
| | | LC81160142015249LGN01 | |

## 4. Results and Discussion

### 4.1. Quantitative Assessment

#### 4.1.1. North America Accuracy Assessment Using NGS Points

To evaluate if the statistics were different over the countries where NGS points were available, we calculated the accuracy of the DEMs over Canada, the Continental United States (CONUS), and Mexico separately (see Figure 5 for distribution). In all regions, CopDEM is less than 1 m (RMSE) more accurate than the Collection-2 DEM (see Table 7 for the accuracy assessment using NGS points). It is interesting to note that for 90% of the points in each region, the difference between the accuracy of the DEMs is roughly 1 m, but when looking at the other 10% (i.e., 95% and 99%) the statistics start to deviate, indicating that the Collection-2 DEM has more outliers than the CopDEM, which could be due to true errors or because the elevation measurements were not collected contemporaneously so the changes could be due to time.

**Table 7.** Accuracy assessment (in meters) using National Geodetic Survey (NGS) points. CONUS, Continental United States; RMSE, root mean square error.

|  | CONUS | | Canada | | Mexico | | North America | |
| --- | --- | --- | --- | --- | --- | --- | --- | --- |
|  | Copernicus | Collection-2 | Copernicus | Collection-2 | Copernicus | Collection-2 | Copernicus | Collection-2 |
| # of Pts | 30,417 | | 570 | | 197 | | 31,185 | |
| Range | −30 to 59 | −25 to 51 | −12 to 11 | −9 to 18 | −18 to 7 | −13 to 6 | −30 to 59 | −25 to 51 |
| Mean | 0.26 | 0.00 | −0.24 | 1.22 | 0.02 | 0.03 | −0.26 | 0.00 |
| Median | −0.12 | −0.23 | −0.16 | 1.06 | 0.07 | 0.12 | −0.12 | −0.21 |
| STD | 1.87 | 2.60 | 2.04 | 2.40 | 1.74 | 2.40 | 1.87 | 2.66 |
| RMSE | 1.90 | 2.63 | 2.07 | 2.68 | 1.75 | 2.41 | 1.90 | 2.66 |
| 90% | 2.76 | 3.80 | 3.07 | 4.23 | 1.61 | 3.48 | 2.76 | 3.83 |
| 95% | 3.91 | 5.36 | 4.57 | 5.17 | 2.47 | 4.72 | 3.91 | 5.40 |
| 99% | 6.72 | 9.24 | 6.51 | 7.97 | 4.89 | 10.67 | 6.72 | 9.43 |

#### 4.1.2. Global Accuracy Assessment Using ICESat-2 Data

To sample the rest of the world, the ICESat-2 dataset was used since it has global coverage and a high density of sample points. To study if the accuracy of the DEMs varied by regions, the statistics were parsed to look at the six major continents. Referencing Table 8, in North America, South America, Europe, Africa, and Australia, the accuracy of both DEMs is very good, ranging from 1.5 (Copernicus in Africa) to 4.5 m (Collection-2 in Europe) RMSE. In all cases except North America, and based only on the RMSE, Copernicus performed slightly better than the Collection-2 DEM, but never by much. This marginal difference in North America, however, is opposite to what was seen when using the NGS points as reference (Table 7, above), where the Copernicus DEM performed slightly better. In the case where ICESat-2 data were used as the reference, the Collection-2 DEM was slightly more accurate. The difference in accuracy between CDEM and ICESat was marginally smaller than that between NASADEM and ICESat over sites in North America (Table 9). This leads us to believe that the accuracy of the CDEM may be marginally better than that of NASADEM and was largely responsible for the better performance of the Collection-2 DEM relative to CopDEM in North America (Table 8). The only statistic in North America where Copernicus did better is that the Median and Absolute Median values are closer together indicating better consistency. All the other metrics, such as RMSE, LE90, and LE95, show lower accuracy which may be due to either artifacts, poor DEM accuracy, or both.

**Table 8.** Accuracy assessment (in meters) using Ice, Cloud, and land Elevation Satellite-2 (ICESat-2) Atlas points, separated by continents. RMSE, root mean square error; LE90, LE95, and LE99 are vertical accuracies at 90, 95, and 90% confidence levels, respectively.

| | North America (18 Sites) | | South America (8 Sites) | | Europe (7 Sites) | | Africa (8 Sites) | |
|---|---|---|---|---|---|---|---|---|
| | Copernicus | Collection-2 | Copernicus | Collection-2 | Copernicus | Collection-2 | Copernicus | Collection-2 |
| # of Pts | 208,094 | | 65,657 | | 93,262 | | 206,364 | |
| Range | −81 to 40 | −81 to 46 | −64 to 21 | −117 to 20 | −40 to 41 | −169 to 126 | −29 to 5 | −21 to 21 |
| Mean | −1.57 | −0.49 | −0.89 | −0.94 | −1.20 | −0.21 | −0.40 | −0.43 |
| Median | −0.40 | −0.04 | −0.06 | −0.16 | 0.15 | 0.12 | −0.03 | −0.29 |
| Abs Median | 0.64 | 1.39 | 0.31 | 1.18 | 0.47 | 1.45 | 0.16 | 1.06 |
| STD | 3.60 | 3.40 | 3.54 | 4.30 | 4.10 | 4.45 | 1.50 | 2.00 |
| RMSE | 3.95 | 3.41 | 3.65 | 4.40 | 4.28 | 4.46 | 1.54 | 2.05 |
| LE90 | 5.58 | 4.56 | 2.75 | 4.37 | 6.11 | 6.64 | 1.20 | 3.11 |
| LE95 | 8.99 | 6.68 | 5.72 | 8.19 | 11.03 | 9.73 | 2.75 | 4.12 |
| LE99 | 16.91 | 12.27 | 20.85 | 20.84 | 18.88 | 15.73 | 6.55 | 6.91 |

| | Asia (12 sites) | | Australia (7 sites) | | All Sites (60 sites) | |
|---|---|---|---|---|---|---|
| | Copernicus | Collection-2 | Copernicus | Collection-2 | Copernicus | Collection-2 |
| # of Pts | 203,210 | | 112,576 | | 889,113 | |
| Range | −98 to 85 | −267 to 393 | −42 to 11 | −42 to 54 | −98 to 85 | −267 to 393 |
| Mean | −0.32 | 0.80 | 0.10 | −0.32 | −0.71 | −0.16 |
| Median | 0.01 | 0.29 | 0.33 | −0.17 | −0.02 | −0.04 |
| Abs Median | 0.36 | 1.23 | 0.41 | 1.08 | 0.35 | 1.20 |
| STD | 1.96 | 8.70 | 1.65 | 2.50 | 2.80 | 5.00 |
| RMSE | 1.98 | 8.72 | 1.65 | 2.51 | 2.90 | 5.04 |
| LE90 | 2.33 | 13.71 | 0.93 | 3.11 | 2.72 | 4.84 |
| LE95 | 4.06 | 20.64 | 1.37 | 3.91 | 5.50 | 8.93 |
| LE99 | 8.15 | 33.54 | 5.10 | 8.54 | 14.13 | 22.88 |

**Table 9.** Accuracy assessment (in meters) using Ice, Cloud, and land Elevation Satellite-2 (ICESat-2) Atlas points for Copernicus and Collection-2 in North America (N.A.), where the Collection-2 source Digital Elevation Model (DEM) statistics were separated. NASADEM, National Aeronautics and Space Administration DEM; CDEM, Canadian DEM; RMSE, root mean square error; LE90, LE95, and LE99 are vertical accuracies at 90, 95, and 90% confidence levels, respectively.

| | Copernicus Stats in N.A. Where Collection-2 Uses | | Collection-2 Stats in N.A. | |
|---|---|---|---|---|
| | NASADEM | CDEM | NASADEM | CDEM |
| # of Pts | 115,463 | 92,631 | 115,463 | 92,631 |
| Range | −59 to 33 | −81 to 40 | −52 to 43 | −81 to 46 |
| Mean | −1.62 | −1.5 | −0.90 | 0.02 |
| Median | −0.29 | −0.62 | −0.37 | 0.29 |
| Abs Median | 0.43 | 0.9 | 1.53 | 1.22 |
| STD | 3.9 | 3.25 | 3.66 | 2.90 |
| RMSE | 4.22 | 3.58 | 3.77 | 2.90 |
| LE90 | 5.57 | 5.58 | 5.27 | 3.62 |
| LE95 | 9.92 | 8.33 | 7.74 | 5.23 |
| LE99 | 18.45 | 13.40 | 13.59 | 10.05 |

When considering all the error metrics, the results are mixed but fall mostly in favor of CopDEM. Europe had mixed results, with the Absolute Mean, STD, RMSE, and LE90 being better with the CopDEM, but the LE95 and LE99 being better with the Collection-2

DEM, indicating that for the most part, Copernicus is more accurate, but with a few outliers influencing the LE95 and LE99. The continents of Asia and Australia had better results in favor of Copernicus, especially in the case of Asia, where 99% of the CopDEM error falls within 8.15 m, whereas 99% of the Collection-2 DEM error falls within 33.54 m

The large discrepancy with the Collection-2 DEM data in Asia is twofold. In the high-relief regions of the Himalayas, NASADEM had many artifacts (leading to errors up to 393 m over tested sites) that affected the quality and accuracy of the DEMs. Additionally, in Northern Russia, the source DEM was never updated since Collection-1 and was using the less accurate GMTED dataset [1,57]. Even with the removal of the egregious points (41 of the 203,210 points had errors greater than 100 m), the statistics were not changed much showing that the error is not simply due to a few large outliers.

Notably, when considering all continents, CopDEM statistics are very consistent compared to the ICESat reference points, with an RMSE in the range of 1.5 to 4 m depending on geographical regions. In comparison, the RSME of the Collection-2 DEM is more variable, fluctuating between 2 and 9 m. The overall statistics for all the sites combined indicate that the RMSE of CopDEM is less than 3 m (Table 8).

In summary, for most continents, the results were similar between the DEMs, except over Asia. Interestingly, where the Collection-2 DEM struggled the most (Asia) was where the CopDEM had some of its better results. Additionally, the mean and median values are mostly similar between the two DEMs, indicating that there are no large outliers affecting the distribution. The CopDEM, primarily being a single dataset source, provides consistent accuracy in the range of 2 to 4 m RMSE globally, which is evident from the statistical distribution. When considering all sites, the CopDEM is roughly 2 m more accurate than the Collection-2 DEM, and 90% of its values have errors less than 3 m and only 1% has errors greater than 14.13 m.

Since the Collection-2 DEM is a seamless mosaic of DEMs from varied sources as shown in Figure 2, the accuracy of the source DEMs used in the Collection-2 DEM was evaluated by comparing with the same ICESat-2 data points extracted from the CopDEM. NASADEM was the largest source for the Collection-2 DEM dataset and was used for 56° South to 60° North providing a consistent DEM at 3 arcseconds resolution. The other datasets were used as a supplement, where NASADEM was not available. Table 10 presents the statistical comparison of different DEM sources used in Collection-2 DEM versus CopDEM. While typically the mean, standard deviation, and RMSE values are the most commonly used metrics for evaluating accuracy, other statistics, such as the range, median, and percentile distribution, are also important. For example, looking at the 46 NASADEM sites, there is only a marginal improvement in the accuracy of Copernicus DEM compared to the Collection-2 DEM; however, the range (i.e., min, max) shows that there must be large artifacts in the Collection-2 dataset. This large range is also evident in the Asia results (Table 8, above) as these tiles lie in the Himalayas. A similar trend is observed for other DEM sources as well. By using percentiles and comparing them to other statistical metrics in the table, it is easier to assess the accuracy of the DEM datasets and the effect of the artifacts on these DEM datasets.

Overall, the accuracy of CopDEM is comparatively better than that of Collection-2 DEM except in Canada, where the difference statistics with respect to ICESat-2 data show that the CDEM source DEM of Collection-2 is marginally better in comparison to the CopDEM (Table 9). The region where GMTED was used in Collection-2 was expected to show the largest improvement because the GMTED dataset was old and never updated from Collection-1. This is evident in Table 10, where the CopDEM has an RMSE of 4.5 m and the Collection-2 DEM (GMTED) is greater than 16 m. The other metrics also support similar observations.

Although CopDEM is not a mosaic of multiple DEM sources like the Collection-2 DEM, the voids in the CopDEM were filled with alternate DEM sources. The accuracy of the filled DEM values was evaluated by comparing the source of the fill DEM with the ICESat-2 data points. This was to learn if the alternate DEM sources chosen to fill the voids

in the Copernicus data were appropriate, or if they could be improved by using Collection-2 DEM sources. Using the Copernicus Fill Mask Layer (Table 4), it was established which points were over voids and they were extracted from the rest of the dataset for comparisons. Since it was a relatively small number of values in comparison to the rest of the dataset, it did not affect the overall statistics much when they were removed; however, when focusing on the accuracy of the fill data alone, the quality became much more apparent (Table 5) when compared to the statistics of the complete datasets (Table 8). Predictably, the accuracy was not as high as the native Copernicus data (being roughly 10–15 m RMSE), but still it was consistent throughout. When compared to those points in the Collection-2 dataset, the CopDEM fill data did not perform as well in four of the six cases, having lower accuracy (RMSE) than the Collection-2 dataset (Table 11). If large artifacts could be removed, then the Collection-2 dataset could be a better choice for fill data (Table 12). Regardless, the analysis of the alternate DEM sources selected to fill the Copernicus voids indicates that if there is a reprocessing of the Copernicus dataset, it would be wise to consider using other newer/better or local DEM sources to improve the voids.

**Table 10.** Accuracy assessment (in meters) using Ice, Cloud, and land Elevation Satellite-2 (ICESat-2) Atlas points, based on the source Digital Elevation Model (DEM) used in Collection-2 DEM. NASADEM, National Aeronautics and Space Administration DEM; CDEM, Canadian DEM; SNF, Sweden–Norway–Finland; GMTED, Global Multiresolution Terrain Elevation Data; RMSE, root mean square error; LE90, LE95, and LE99 are vertical accuracies at 90, 95, and 90% confidence levels, respectively.

| | NASADEM (46 Sites) | | CDEM (8 Sites) | | SNF (1 Site) | | ArcticDEM (2 Sites) | | GMTED (3 Sites) | |
|---|---|---|---|---|---|---|---|---|---|---|
| | Copernicus | Collection-2 | Copernicus | Collection-2 | Copernicus | Collection-2 | Copernicus | Collection-2 | Copernicus | Collection-2 |
| **# of Pts** | 672,299 | | 92,631 | | 18,632 | | 51,548 | | 54,003 | |
| **Range** | −98 to 85 | −117 to 393 | −81 to 40 | −81 to 46 | −8 to 14 | −28 to 22 | −16 to 17 | −169 to 126 | −29 to 14 | −267 to 120 |
| **Mean** | −0.60 | −0.53 | −1.50 | 0.02 | 0.47 | −0.48 | −0.20 | 0.38 | −1.70 | 3.70 |
| **Median** | 0.01 | −0.19 | −0.62 | 0.29 | 0.45 | −0.50 | 0.40 | 0.75 | −0.29 | 3.70 |
| **Abs Median** | 0.28 | 1.11 | 0.90 | 1.22 | 0.62 | 1.74 | 0.92 | 1.10 | 0.53 | 9.60 |
| **STD** | 2.64 | 3.00 | 3.25 | 2.90 | 1.18 | 3.40 | 2.06 | 4.00 | 4.10 | 15.90 |
| **RMSE** | 2.70 | 3.07 | 3.58 | 2.90 | 1.26 | 3.46 | 2.07 | 4.02 | 4.46 | 16.32 |
| **LE90** | 2.06 | 3.59 | 5.57 | 3.62 | 1.58 | 5.73 | 3.22 | 3.72 | 4.96 | 23.07 |
| **LE95** | 4.41 | 5.25 | 8.83 | 5.23 | 2.08 | 7.31 | 4.77 | 6.79 | 11.64 | 31.03 |
| **LE99** | 13.85 | 11.97 | 13.40 | 10.05 | 4.62 | 10.99 | 7.67 | 15.52 | 18.50 | 46.06 |

*4.2. Qualitative Assessment*

4.2.1. Anaglyphs Created South of 60°N. Latitude

The quality of the anaglyph images of the three assessed regions showed mixed results in terms of their DEM accuracy and quality. In Austria, there were mixed results with misalignments in both processed image pairs. In the Himalayas along the China/Nepal border, CopDEM performed better and in the Himalayan tile in Pakistan, the images processed using the Collection-2 DEM performed better. For the image pairs over Austria (WRS-2 Path 192, Row 27 and Path 193, Row 27), for example, there were registration errors caused by both DEMs in the same region. As shown in the left panel of Figure 14, Collection-2 DEM did a better job aligning image pairs for ridges (grayscale) than for the base of a nearby mountain face (circled area). The Copernicus quality bands show that in the upper region where the misalignment was observed (circled in the right panel of Figure 14), part of the region was edited by smoothing, but in the lower region where the CopDEM performed better (grayscale), the region was filled with Advanced Spaceborne Thermal Emission and Reflection Radiometer ASTER DEM data (see Figure 15). In the Himalayan region near the China/Nepal border (WRS-2 145/39 and 146/39), misalignment errors can be visually observed (note the spread of red color) in the anaglyph processed using Collection-2 DEM,

but the same regions were free of any misalignments in the CopDEM (Figure 16). In this tile, the Copernicus Fill Mask (not shown) indicated that this area was filled using SRTM30 data. In contrast, the anaglyph created in the northern Himalayan region in Pakistan showed superior performance by the Collection-2 DEM in some of the most challenging terrains (Figure 17). In this case, when referencing the Copernicus Fill Mask (not shown), it indicated that the artifact region was filled with SRTM30+ data. While creating the anaglyphs over these regions, results show flaws in both datasets; flaws showed up equally and neither dataset was superior to the other. Seemingly, this is a surprising result since the CopDEM has higher spatial resolution than the Collection-2 DEM and should therefore perform better in these mountainous regions because it is better suited to capture steep terrain changes. However, this also shows that higher-resolution data do not always lead to better results. DEM accuracy is also necessary to achieve accurate registration. Additionally, as explained above, in all three of these examples, the CopDEM used alternate sources to fill data voids. Therefore, it is not surprising that this is where the misalignments occurred, especially since there must have been some interpolation and smoothing performed to blend the fill data.

**Table 11.** Accuracy (in meters) of Digital Elevation Model (DEM) values used to fill voids in the Copernicus DEM (CopDEM). RMSE, root mean square error; LE90, LE95, and LE99 are vertical accuracies at 90, 95, and 90% confidence levels, respectively.

| | North America | | South America | | Europe | | Africa | |
|---|---|---|---|---|---|---|---|---|
| | Copernicus | Collection-2 | Copernicus | Collection-2 | Copernicus | Collection-2 | Copernicus | Collection-2 |
| # of Pts | 858 | | 125 | | 511 | | 21 | |
| Range | −53 to 17 | −43 to 29 | −44 to 4 | −52 to 12 | −32 to 41 | −50 to 62 | −22 to 3 | −17 to 6 |
| Mean | −9.90 | −8.40 | −8.10 | −10.20 | −7.90 | −5.40 | −6.50 | −5.90 |
| Median Abs | −8.90 | −7.60 | −4.90 | −8.00 | −8.00 | −5.20 | −4.10 | −6.60 |
| Median | 8.90 | 7.70 | 4.90 | 8.30 | 8.60 | 6.30 | 4.10 | 6.60 |
| STD | 7.70 | 7.30 | 10.00 | 10.90 | 9.70 | 8.60 | 7.20 | 6.20 |
| RMSE | 12.49 | 11.16 | 12.88 | 14.87 | 12.49 | 10.16 | 9.59 | 8.44 |
| LE90 | 18.18 | 15.20 | 20.98 | 26.32 | 20.79 | 14.38 | 14.59 | 12.76 |
| LE95 | 21.58 | 22.21 | 26.77 | 28.31 | 22.89 | 16.46 | 21.09 | 12.78 |
| LE99 | 41.16 | 38.38 | 42.31 | 45.37 | 28.99 | 32.64 | 22.21 | 16.67 |
| | Asia | | Australia | | Global | | | |
| | Copernicus | Collection-2 | Copernicus | Collection-2 | Copernicus | | Collection-2 | |
| # of Pts | 529 | | 137 | | 2181 | | | |
| Range | −98 to 85 | −116 to 393 | −42 to 4 | −42 to 6 | −98 to 85 | | −116 to 393 | |
| Mean | −0.50 | −0.82 | −10.40 | −4.70 | −7.00 | | −5.70 | |
| Median Abs | 0.08 | −0.05 | −1.77 | −3.27 | 10.90 | | −5.80 | |
| Median | 4.50 | 5.40 | 2.60 | 3.40 | 7.40 | | 6.80 | |
| STD | 13.20 | 27.90 | 13.50 | 7.10 | 10.90 | | 15.70 | |
| RMSE | 13.20 | 27.89 | 16.97 | 8.46 | 12.98 | | 16.73 | |
| LE90 | 18.40 | 21.64 | 30.28 | 11.61 | 20.13 | | 16.56 | |
| LE95 | 26.71 | 33.29 | 33.93 | 14.12 | 25.72 | | 24.47 | |
| LE99 | 52.31 | 90.48 | 38.50 | 41.07 | 41.24 | | 42.64 | |

**Table 12.** Fill statistics in Asia with 4 egregious (i.e., error > 100 m) points removed for an assessment of the accuracy (in meters) of Digital Elevation Model (DEM) values used to fill voids in the Copernicus DEM (CopDEM). RMSE, root mean square error; LE90, LE95, and LE99 are vertical accuracies at 90, 95, and 90% confidence levels, respectively.

| | Asia | | Global | |
|---|---|---|---|---|
| | **Copernicus** | **Collection-2** | **Copernicus** | **Collection-2** |
| **# of Pts** | 525 | | 2177 | |
| **Range** | −98 to 85 | −96 to 91 | −98 to 85 | −96 to 91 |
| **Mean** | −0.32 | −1.75 | −7.00 | −5.90 |
| **Median** | 0.08 | −0.05 | −6.40 | −5.80 |
| **Abs Median** | 4.50 | 5.40 | 7.40 | 6.70 |
| **STD** | 12.90 | 15.70 | 10.90 | 10.80 |
| **RMSE** | 13.20 | 15.90 | 12.98 | 12.31 |
| **LE90** | 17.54 | 21.05 | 19.99 | 16.37 |
| **LE95** | 25.74 | 28.17 | 25.20 | 23.95 |
| **LE99** | 52.31 | 73.58 | 41.16 | 41.07 |

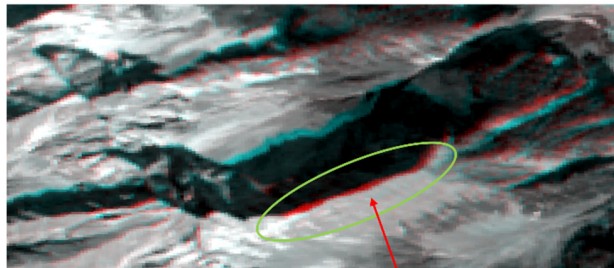
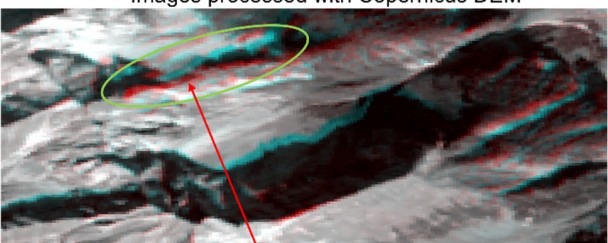

**Figure 14.** Anaglyphs created in Austria (Landsat Worldwide Reference System-2 (WRS-2 192/27 and 193/27)) showing varying quality. The left image, processed using the Collection-2 Digital Elevation Model (DEM), shows misalignment in the lower circled region, while the right image, processed using the Copernicus DEM (CopDEM), shows misalignment just north of it. Color variations (red and cyan) indicate images that are misaligned, while grayscale indicates images that are perfectly aligned.

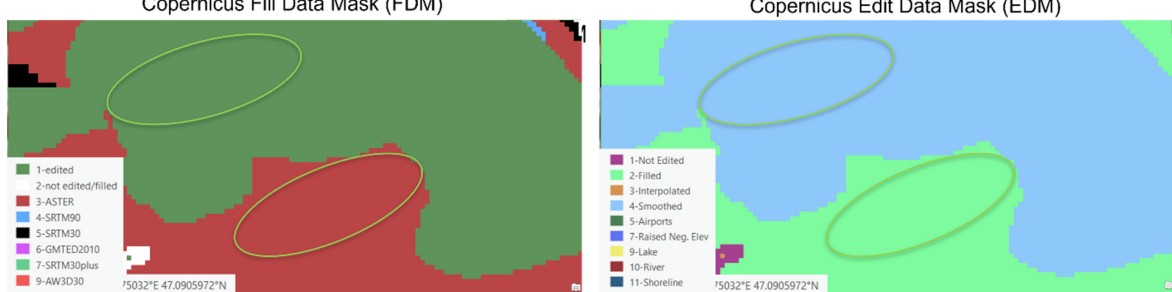

**Figure 15.** Voids edited by the Copernicus Digital Elevation Model (CopDEM). Panels depict FDM and EDM quality layers for a tile in the same region in Austria (WRS-2 192/27 and 193/27) as shown in Figure 14. In the right image, where the EDM shows the pixels were smoothed (upper circle) the CopDEM produced misalignment in the processed image (Figure 14, right image, upper circle). In the left image, where the FDM shows the pixels were filled with Advanced Spaceborne Thermal Emission and Reflection Radiometer (ASTER) DEM data (lower circle), there were no misalignments of the processed image evident in the anaglyph (Figure 14, lower region of right image). SRTM90, Shuttle Radar Topography Mission 90 m resolution; SRTM30, SRTM 30 m resolution; GMTED2010, Global Multiresolution Terrain Elevation Data 2010.

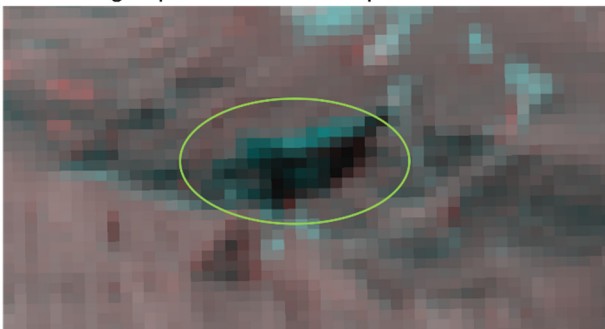

**Figure 16.** Anaglyph in the Himalayan region along the China/Nepal border where there is more misalignment (indicated by the spread of red color) when processing the imagery using the Collection-2 DEM (**left**) which is not evident in the imagery processed using the Copernicus DEM (CopDEM) (**right**).

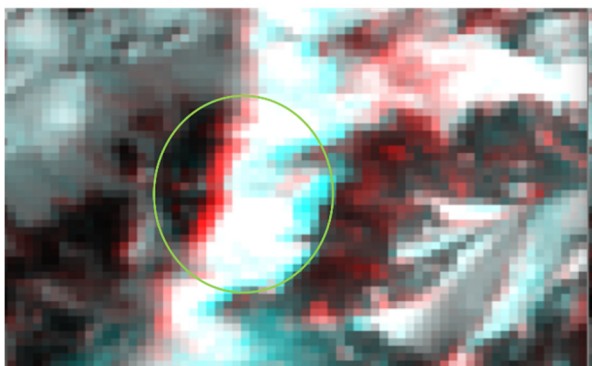
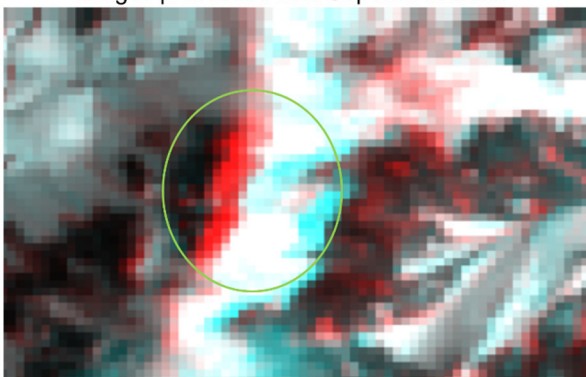

**Figure 17.** Anaglyph in the Pakistan region of the Himalayas where there is more misalignment when processing the imagery using the Copernicus Digital Elevation Model (CopDEM) (**right**) which is not as prevalent (less red and blue) in the imagery processed using the Collection-2 DEM (**left**).

Processing using the 1 arcsecond CopDEM did not show notable improvement over using the Collection-2 DEM at 3 arcseconds. To further explore this, we then used the 1-arcsecond NASADEM version instead of the 3-arcsecond version that was used in Collection-2 to see if it improved some of the misalignments. The misalignments did not improve, confirming the conclusions of other researchers that dataset accuracy is more important than dataset resolution [58–61]. Although the anaglyph comparisons did not reveal any conclusive evidence on the quality and accuracy of the DEM in these high-relief regions, one issue it did expose is the number of void pixels that were filled in the CopDEM with alternate DEM sources in these regions (see below, the section on fill analysis—Section 4.2.3). Note that what is shown above are just a few representative examples of what was observed when comparing the anaglyphs of these image pairs. Many more errors were observed in these regions, and it was unclear as to which of the two DEM datasets (Collection-2 DEM and CopDEM) provides more accurate elevation data.

### 4.2.2. Anaglyphs Created North of 60°N. Latitude

The regions that were tested north of the 60°N latitude are shown in Figure 13. The region in Scandinavia that was studied was along the Sweden/Norway border, where the Collection-2 dataset used the SNF DEM as its source. The anaglyph did not reveal any improvements when using the CopDEM; however, when comparing the two datasets, there were differences up to 335 m (Table 13). It cannot be definitively determined which DEM was the inaccurate one since there were no ICESat-2 reference points that fell exactly over

the specific pixels, but we expect that the error is most likely due to the void fill using the ASTER DEM since those large differences were located exactly where there were voids. Likewise, the anaglyph created over Iceland also did not show any improvements using the newer CopDEM. The statistics are similar to the SNF example, but in this case, the differences were mostly due to the extensive void-filling process required to complete the ArcticDEM. It is interesting to note that although differences were found between the Collection-2 and CopDEM datasets, they did not introduce misalignments in the anaglyphs, denoting that the differences in elevation between the datasets are not large or systematic enough to make any noticeable improvement in the geolocation of the processed images. The standard deviation in Sweden/Norway and Iceland regions suggests that there are systematic differences due to either mis-registrations, resolution differences, or a combination of both. In addition, the differences in the range suggest that outliers exist in one of the datasets.

**Table 13.** Digital Elevation Model (DEM) difference statistics (in meters) between Copernicus DEM and Landsat Collection-2 DEM.

| Location | Minimum | Maximum | Mean | STD |
|---|---|---|---|---|
| Sweden/Norway | −335 | 282 | 0.21 | 6.81 |
| Iceland | −474 | 243 | 0.19 | 5.16 |
| Northern Russia | −263 | 563 | 6.02 | 25.65 |

A different result was found for the study site in northern Russia when the anaglyphs were made. Not only were the difference statistics much larger between the datasets (see Table 13, above), but the anaglyph created using the CopDEM showed better alignment than the one created using the Collection-2 DEM (see Figure 18). While in the other locations where the anaglyphs were made the difference in spatial resolution did not affect the image registration, the source DEM for the Collection-2 dataset in Northern Russia was GMTED, which was collected at 7.5-arcsecond resolution [5], being much lower than the CopDEM (see Figure 19). In this case, we believe that the errors introduced in the elevation by GMTED are likely due to a combination of the low native resolution of 7.5 arcseconds and less accurate dataset of GMTED, as was demonstrated when comparing it to the ICESat-2 points (Table 10). This is evident in the anaglyph images as misalignments over peaks, as shown in Figure 18.

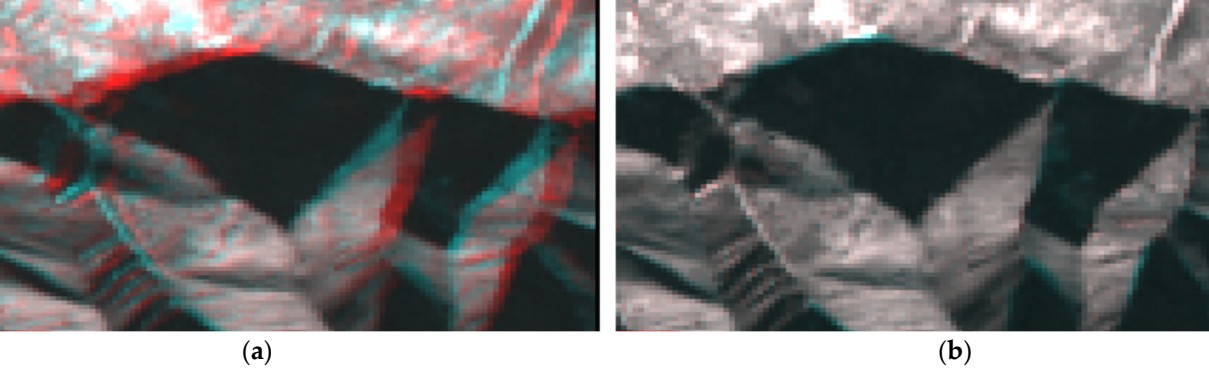

(**a**)　　　　　　　　　　　　　　　　(**b**)

**Figure 18.** Anaglyph made using the Collection-2 Digital Elevation Model (DEM) showing significant misalignments (indicated by the red and cyan) on the mountain ridges (**a**), and the anaglyph made with the Copernicus DEM (CopDEM) having none (**b**).

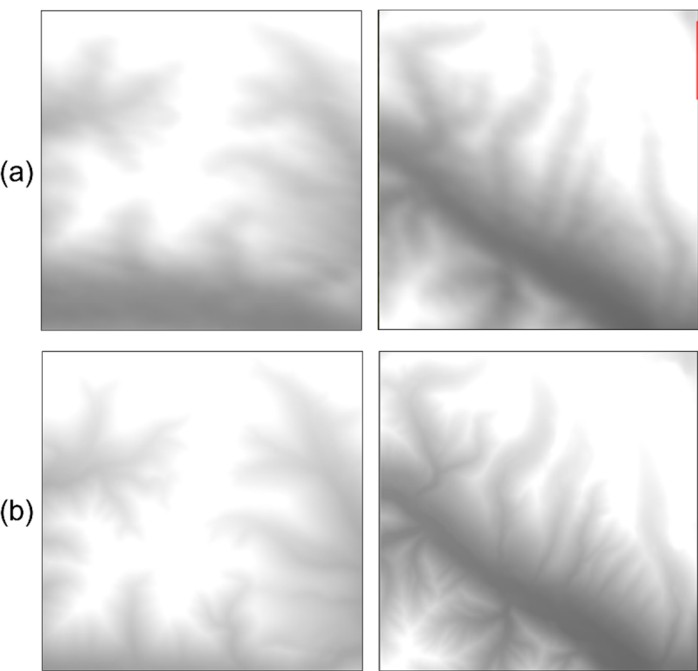

**Figure 19.** The top row shows two locations in the Collection-2 Digital Elevation Model (DEM) where a 7.5-arcsecond Global Multiresolution Terrain Elevation Data (GMTED) was the source DEM (**a**), and where the same locations in the GLO–30 Copernicus DEM (CopDEM) were the source DEM (**b**). The difference in sharpness due to the increased spatial resolution of the CopDEM is evident.

### 4.2.3. Fill Analysis of CopDEM

While globally the CopDEM has only a very small percentage of its data coming from another source, in the mountainous regions, the percentage was much higher. This data distribution agrees with results found in other studies, albeit the percentage in the mountainous regions in those studies is not as high as in the tiles we were analyzing [31]. In the three regions that were analyzed south of 60°, unedited CopDEM data varied from 42% to 87% (Figure 20). This shows that in some challenging terrains, the number of filled pixels can be greater than 50% of the tile. For example, in the northern Himalayas, native Copernicus data were not the majority source, but fill data DEMs were dominant, equaling 56% of the tile when adding together the five alternate DEM sources. Of the fill sources, SRTM30 and SRTM30+ were used most often (see Table 4 for more explanation of the fill sources).

The most challenging places for synthetic aperture radar (SAR) interferometry to collect elevation data are dense forests, highly built-up, or mountainous regions. The amount of fill we observed in our analysis is higher than what was published in [31], where the authors found up to 26% of the data filled with alternate DEM sources. Although a large percentage of CopDEM data were filled with alternate DEM sources in the sites we analyzed, our results show that the elevation accuracy did not degrade very much, with accuracies comparable to the accuracy of the Collection-2 DEM in those regions. As noted above, the largest fill sources for our study area were SRTM derivatives. SRTM date were collected significantly earlier (early 2000) than TanDEM-X data (i.e., the source for CopDEM), which were collected between 2010 and 2015, and consequently, it is necessary to bear this fact in mind as it also means that the suitability of the CopDEMs for multitemporal analysis is limited if the region has experienced a lot of elevation changes with respect to time and if a lot of fill imagery was used. As such, if temporal consistency is necessary, users should consult auxiliary data such as the filling mask before conducting any analyses to learn if there were significant data voids and what alternate DEMs were used [29,62].

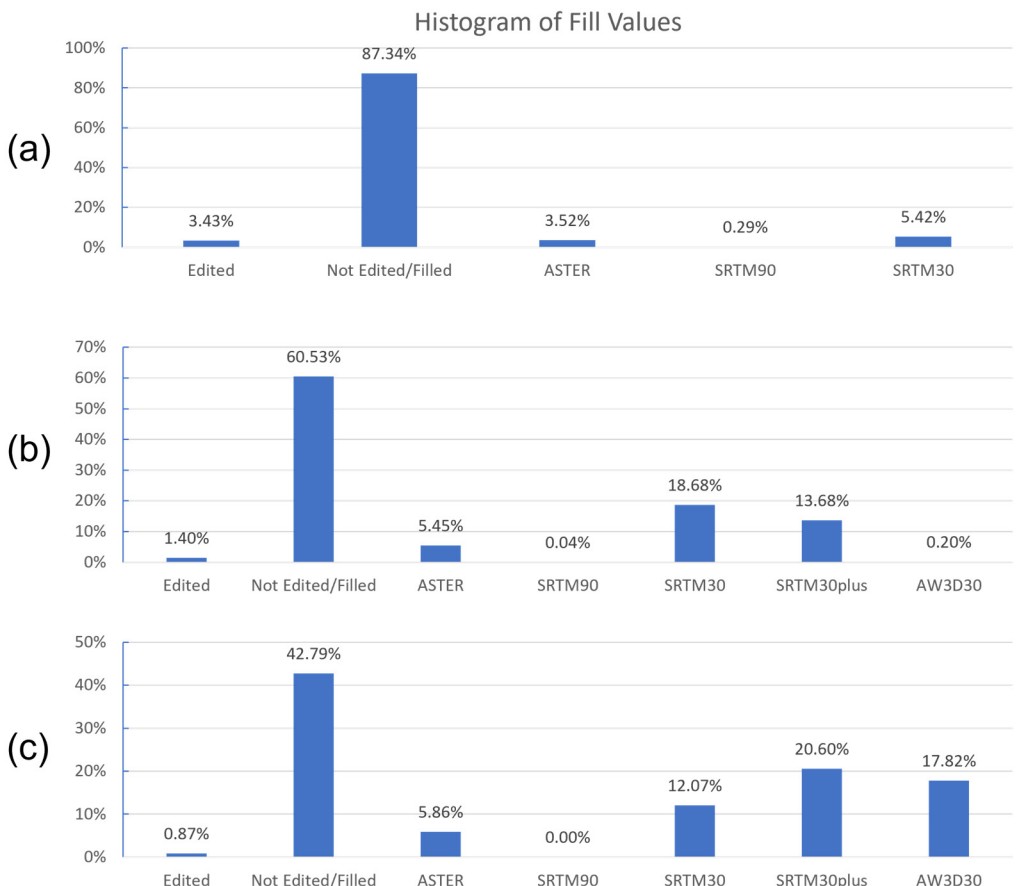

**Figure 20.** Histogram of Copernicus pixels in the Fill Layer Mask (FLM) tiles in Austria (**a**), Central Himalayas (**b**), and Northern Himalayas (**c**) showing the percentage of Copernicus-edited pixels, unedited pixels, and alternate Digital Elevation Model (DEM) sources used. ASTER, Advanced Spaceborne Thermal Emission and Reflection Radiometer; SRTM90, 30, and 30plus, Shuttle Radar Topography Mission for 90, 30, and 30 plus meter resolution; AW3D30,(ALOS World 3D-30 m).

## 5. Conclusions

Currently, the USGS uses their Collection-2 DEM which is a mosaic of several elevation datasets with varying quality, accuracies, and collection timeframes resampled to a 3-arcsecond spatial sampling. The recently released CopDEM, produced by the European Union's Copernicus Earth Observation program, is publicly available, has global coverage at the 1-arcsecond spatial resolution and, based on our observation, is virtually seamless with a very small amount of void fill data, except in regions of high relief. The goal of this work was to determine if using the CopDEM for terrain correction would improve the registration accuracy of Landsat imagery. To verify the improvements, we compared the two DEMs using quantitative and qualitative methods. Quantitatively, using NGS and ICESat-2 points as control, the difference between the two DEMs was less than 1-meter RMSE in all regions except Asia, where the Collection-2 DEM is not as accurate in the high-relief regions such as the Himalayas and in northern Russia where Collection-2 still was using a coarsely sampled GMTED DEM. It is worth noting that the varying differences in accuracy are due more to the fluctuation of Collection-2 DEM accuracy, which ranged from 2 m to 9 m RMSE, than to the CopDEM, which had more consistent global accuracy, ranging from 1.5 m to 4.28 m RMSE. This is undoubtedly due to the Collection-2 DEM having different source DEMs, whereas the CopDEM has mainly one source with other sources only being used to fill voids.

From a qualitative perspective, apart from one region where the Collection-2 DEM used the GMTED dataset, the anaglyph results obtained using both DEM datasets were

somewhat inconsistent between different regions, and therefore, it was not easy to decide which of the two DEM datasets overall was significantly better. Although our quantitative assessment shows differences, the anaglyphs highlighted that except in the case of GMTED, the differences were not large enough to translate to a corresponding horizontal misalignment that showed up in the processed imagery.

As an additional analysis, the fill data used for the Copernicus DEM were compared to values for the same points extracted from the Collection-2 DEM. This was performed to learn if the Copernicus program could have executed better by using Collection-2 data for those regions rather than other source imagery. In most cases, the accuracy of the fill data used by Copernicus was similar to that of data extracted from the Collection-2 DEM, except for cases in Asia and Australia. In Asia, the accuracy of the fill data used by Collection-2 DEM was far worse than that of the fill imagery used by the Copernicus program, although this was due mainly to 4 egregious points in the Himalayas (Table 12). In contrast, the accuracy of the Collection-2 data for Australia was much better than that of the Copernicus data. This was attributed more to the lower accuracy of the fill data used for the Copernicus DEM than to the performance of the Collection-2 DEM. This was surprising since in the Australian region the terrain is relatively flat (except for mountains in southern New Zealand), and the same fill sources were used (SRTM30 and SRTM90) as in other places. However, since there were so few sample data points (137), the result is not highly reliable.

Overall, the CopDEM has better accuracy when compared to ICESat and NGS datasets and is likely to be more reliable, in general, based on our analysis. The largest and most significant benefit of using the CopDEM was found to be in northern Russia, where both the quantitative and qualitative improvements are stark, due to the Collection-2 DEM using the inaccurate and coarser-resolution GMTED dataset as its source. It is the intention of the Landsat Cal/Val team to use the CopDEM in Collection-3 processing, which will be released no sooner than 2025.

**Author Contributions:** Conceptualization, S.F. and R.R.; methodology, S.F. and R.R.; software, S.F. and R.R.; formal analysis, S.F. and R.R.; investigation, S.F. and R.R.; resources, S.F.; data curation, S.F.; writing—original draft preparation, S.F.; writing—review and editing, S.F. and R.R.; supervision, R.R.; project administration, R.R. All authors have read and agreed to the published version of the manuscript.

**Funding:** This research received no external funding.

**Data Availability Statement:** No new data were created or analyzed in this study. Data sharing is not applicable to this article.

**Acknowledgments:** We received the National Geospatial Survey (NGS) Points from Dean Gesch, at the USGS.

**Conflicts of Interest:** The authors declare no conflict of interest.

## Abbreviations

| | |
|---|---|
| ASTER | Advanced Spaceborne Thermal Emission and Reflection Radiometer |
| Cal/ValCDEM | Calibration and ValidationCanadian Digital Elevation Model |
| CONUSDEM | Continental United States Digital Elevation Model |
| DGEDDSM | Defense Gridded Elevation Data Digital Surface Model |
| DTED | Digital Terrain Elevation Data |
| DTM | Digital Terrain Model |
| EGM | Earth Gravity Model |
| ESAESRI | European Space AgencyEnvironmental Systems Research Institute |
| GDEM | Global Digital Elevation Model |
| GLS | Global Land Survey |

| GMTED2010 | Global Multiresolution Terrain Elevation Data 2010 |
| --- | --- |
| ICESat | Ice, Cloud, and Land Elevation Satellite |
| NASA | National Aeronautics and Space Administration |
| NED | National Elevation Dataset |
| NGSNPI | National Geodetic Survey Norwegian Polar Institute |
| RAMP | Radarsat Antarctic Mapping Project |
| RMSE | Root Mean Square Error |
| SNFSRTM | Sweden–Norway–Finland DEMShuttle Radar Topography Mission |
| STD | Standard Deviation |
| USGS | U.S. Geological Survey |

Any use of trade, firm, or product names is for descriptive purposes only and does not imply endorsement by the U.S. Government.

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
