# Peer review of "Evaluation of Copernicus DEM and Comparison to the DEM Used for Landsat Collection-2 Processing"

_remotesensing, doi:10.3390/rs15102509_

Round 1

Reviewer 1 Report

Summary

This is an overall well-drafted paper. It tells a compelling story from laying out background that introduces the DEM data to be compared, to what methods are used to compare and why to use those methods, all the way to carrying out results and discussions in detail. I enjoyed reading the paper very much and have no trouble understanding the methodology and discussions. I think the main contribution of this paper is that it thoroughly compared CopDEM and Collection-2 DEM from both quantitative and qualitative perspectives and showed that CopDEM has overall better accuracy than Collection-2 DEM. With that, I highly recommend to accept this paper with or without the suggested edits below.

Specific comments

Minor suggestion: Line 317 – 320 may be more readable if those parameter values are shown in a table.

Minor suggestion: Would it be better to rotate Table 7 and Table 10 such that they show as one full figure?

Line 321 – 325: font seems different.

Figure 12: The longitude and latitude degrees are barely recognizable.

Table 7: Would be better if units are specified somewhere.

Reviewer 2 Report

Congratulations on a very substantial study and a generally well-written paper.  It could be published (almost) as is.  But here are my perspectives on relevant issues.

Studies that try to demonstrate “which dataset is better” often rely too much upon sampled statistics.  In doing so they often find “which dataset is usually better”.  A dataset with finer detail is usually better.  But if (elsewhere) it has large glitches, and the alternative dataset is not so bad, then the dataset which is “usually not better” might be better overall.  You intend the selected DEM to be used globally for orthorectifying Landsat.  But you did not do a “find all the problems” global study.  Where exactly does the Copernicus DEM have problems and how bad are they?  And where are there problems in alternative DEMs and how bad are they?  You apparently took some looks here and there.  But it is difficult to draw global conclusions from scattered looks.

Here is how you could do a global search for glitches.  Automatically produce shaded-relief anaglyphs for every quad.  (Takes computer time, very little human time.)  Then look at every quad in 3D.  (Takes human time but typically only seconds per quad for most quads.). Each quad can be flashed upon the screen with the click of a button (easy on a Mac).  I have looked at every quad of NASADEM (15,000+ quads) and ASTER GDEM (22,000+ quads) this way.  Tour the entire world in 3-D!  Informative, interesting, and even fun.  Errors are obvious.  DEM errors seldom look like natural topography.

You could compile a list and map of known DEM (and thus rectification) errors.  This would be very useful to Landsat users.

Copernicus has obvious errors in the Himalayas at 29.5N  85.1E, and there are likely other such notable errors.  What is your concern about the impact of these errors on Landsat rectification?  

Apparently, based on this study, the Copernicus DEM is already the planned DEM for Landsat Collection-3 processing.   If you add a global glitch survey, then might that change?

Do your limited sample site statistics even address the main issue?  Will users care more about RMSE errors of 3m versus 5m somewhere, or will they care more about errors of tens or hundreds of meters somewhere else?  Are you sure that the hundreds-of-meters errors are less common in Copernicus than in NASADEM?  How would you know if you didn’t do a global search?

The anaglyph method you did use apparently only tested six sites:  for the entire world.  It’s a rational method, but six sites are way too inadequate to contribute much to a global DEM evaluation.  It just demonstrated a technique. 

You prefer Copernicus because it is “consistent” (not counting the void fill).  But is consistent preferred to locally or regionally better?  Are errors in Wyoming okay because they are from data “consistent” with the DEM data type used in Vietnam?  Explain why globally consistent is now preferred to “better at each site”.

Line 244:  Change to “Since the Collection-2 DEM used many different source DEMs there was…”

(Otherwise it sounds like you are still talking about NASADEM from the previous sentence.)

Line: 366:  ROI not defined.  (yeah, Region of Interest).

Line 407:  “then” not “than”

Line 438:  You say “other source DEMs”.  

   This would make sense if you made Line 410:

   “3.2.1 South of 60N latitude: NADADEM, focusing on regions of high relief”

   (Other than what?  Other than NASADEM.)

   (Subsection 3.2.2 heading lists dataset names.  3.2.1 does not.)

Line 564:  “affect” not “effect”

Line 702:  In some areas “the number of filled pixels can be greater than 50% of the tile.”

BUT Lines 733-734 say: Cop DEM has “very small amount of void fill.”

WHICH IS IT?   Perhaps “Very little except in xxxxx terrain.”

Figure 15:  Shows nothing is obvious.  Please fix.

Figures 15 & 16 & 17:  You made blue-versus-yellow anaglyphs instead of cyan-versus-red anaglyphs.  Oops.  Nobody has blue-and-yellow glasses.  Please fix.

Line 371:  “These shifts are best viewed using 3D glasses.”  (Yes, I would like to.)

Line 403:  “Set the Green and Blue bands to the same scene and Red to be the other scene”  YES, DO THAT.  YOU DID NOT DO THAT.  You set red and green (yellow) to the same scene and blue to the other scene.

Reference 2 is missing some authors.

CHECK ALL YOUR REFERENCE CITATIONS.  

DO NOT EXPECT REVIEWERS TO CHECK THEM ALL FOR YOU.

DO NOT IMPOSE REFERENCE ERRORS ON YOUR READERS.

Did you do adequate (or any?) statistics comparing NASADEM 1-arcsec to NASADEM 3-arcsec, and >>>both<<< to NGS and Icesat?  How much (if any) superiority of Copernicus (1-arcsec) over NASADEM (at 3-arcsec) is due to the differing “resolutions”.  You tried NASADEM 1-arcsec in your qualitative anaglyph work.  Would be much more interesting in your quantitative work.  If the Collection-2 DEM had been produced at 1-arcsec and used NASADEM 1-arcsec , would Tables 6, 7, 8, and 9 look different, even (perhaps) altering your paper’s main conclusion?

Be careful with the term “resolution”.  It does not mean pixel size or pixel spacing.  What is the resolution of a blurry picture?  Not the pixel size.   In a blurry picture you cannot “resolve” things the size of a pixel.  Most images, and DEMs, are oversampled to preserve the system’s inherent resolution. (Yes, I know that the general public has corrupted the term “resolution” with the advent of personal digital cameras.  But this is a scientific paper.  Be scientific here.  Pixels are NOT resolution.)

Is using “Copernicus everywhere” desirable largely because it is easier (which you knew from the start).  Easy is an advantage, and close enough can be okay.

I appreciate that you are fair several times with your observations that Copernicus is not always better and is often not much better, even statistically in glitch-free locations.

Reviewer 3 Report

This study conducted quantitative and qualitative comparisons between the CopDEM and the Landsat Collection-2 DEM. However, the significance of comparing the two DEM datasets is not fully stated in this paper. The author needs to highlight the contribution of this paper. In addition, there are some points regarding the accuracy assessment results that must be clarified before publication can be considered.

1. Introduction

I suggest that the author introduces other studies related to the evaluation of DEM datasets in the ‘Introduction’ and highlights the innovation and significance of this study.

2. Datasets used in study

Section 2 is too long. I suggest the author further condense the description of the DEM datasets.

3. Methodology

When conducting qualitative evaluation, the study areas selected by the authors are all located near 60 degrees north latitude. I hope the authors can explain why there are no study areas in the southern hemisphere.

4.1.2. Global accuracy assessment using ICESat-2 data

I suggest that the author use figures to show the spatial distribution of research sites on the six continents.

It can be seen from Table 6 that in North America, the root mean square error of CopDEM is lower than that of Collection-2 DEM. However, from Table 7, it can be seen that in North America, the root mean square error of CopDEM is higher than that of Collection-2 DEM. Why is this the case?

On line 277, the author mentioned that 'it is important to note that the elevation error in ICESat data increases with increasing slope and vegetation cover'. Therefore, when comparing the accuracy of CopDEM and Collection-2 DEM using ICESat data as a reference on different continents, would the error of ICESat data lead to erroneous comparison results?

I recommend that the author carefully proofread the format of the manuscript, as there are many unnecessary spaces in the text, as well as a font error in line 322.

Reviewer 4 Report

The manuscript presents Evaluation of Copernicus DEM and comparison to the DEM 2 used for Landsat Collection-2 processing. Further development is in the terms of comprehensive knowledgebase for comparison DEM from available data sources is still necessary.

The article needs to be modified and explained.

Recommendations for addition:

First of all, I would like to congratulate the authors for their work and dedication in carrying out this study.

1.       Abstract: in this part of study, it is necessary to specify in more detail the achieved results of the comparison study, the current form of the abstract is too general and the reader does not have the opportunity to learn more about results of this study.

2.       The main section 1. Introduction is supposed to be an overview of the current state of knowledge linked in the given topic, based on professional studies of the given kind, this part has insufficient links to works of similar importance. The recommendation to expand and supplement with an emphasis on highlighting what makes the given study different and what new knowledge this study brings with its research.

3.       The main section 2. Datasets used in study, some subsections are poor in terms of content, for the study of the given meaning, it is required to expand their content by other relevant facts, otherwise they do not fulfill their meaning (2.1.4. Copernicus DEM Accuracy, 2.3 National Geodetic Survey (NGS), 2.4 Ice, Cloud, and land Elevation Satellite (ICESat).

4.       The main section 3. Methodology – in the introductory part of this chapter, I recommend the authors to supplement the flow chart, which will be a generalization of the methodology used in this study.

5.       The main section 4. Results and Discussion – the section fails to show how this study's results corroborate or differ from prior studies and the likely explanations. There is very less of literature cited in the section.

6.       The study needs to be major revised/reworked, added.

Round 2

Reviewer 3 Report

Thank you for your carefully response. I have no furhter problem.

Reviewer 4 Report

Thanks to the authors for revising and supplementing study based on my comments from last time. All comments have been incorporated. The article has been reworked in very large scale, which significantly increased the quality of scientific processing of the article.